# Rethinking Robustness in Machine Learning: A Posterior Agreement Approach

**João B. S. Carvalho**[*]                                           *joao.carvalho@inf.ethz.ch*
*Department of Computer Science, ETH Zurich*

**Víctor Jiménez Rodríguez**[*]                    *victor.jimenezrodriguez@inf.ethz.ch*
*Department of Computer Science, ETH Zurich*

**Alessandro Torcinovich**[*][†]                     *alessandro.torcinovich@inf.ethz.ch*
*Faculty of Engineering, Free University of Bozen-Bolzano*
*Department of Computer Science, ETH Zurich*

**Antonio E. Cinà**                                              *antonio.cina@unige.it*
*Department of Computer Science, University of Genoa*

**Carlos Cotrini**                                               *ccarlos@inf.ethz.ch*
*Department of Computer Science, ETH Zurich*

**Lea Schönherr**                                              *schoenherr@cispa.de*
*CISPA Helmholtz Center for Information Security*

**Joachim M. Buhmann**                                          *jbuhmann@inf.ethz.ch*
*Department of Computer Science, ETH Zurich*

**Reviewed on OpenReview:** *https: // openreview. net/ forum? id=Bpc9uZ6kcg*

## Abstract

The robustness of algorithms against covariate shifts is a fundamental problem with critical implications for the deployment of machine learning algorithms in the real world. Current evaluation methods predominantly measure robustness through the lens of standard generalization, relying on task performance measures like accuracy. This approach lacks a theoretical justification and underscores the need for a principled foundation of robustness assessment under distribution shifts. In this work, we set the desiderata for a robustness measure, and we propose a novel principled framework for the robustness assessment problem that directly follows the Posterior Agreement (PA) theory of model validation. Specifically, we extend the PA framework to the covariate shift setting and propose a measure for robustness evaluation. We assess the soundness of our measure in controlled environments and through an empirical robustness analysis in two different covariate shift scenarios: adversarial learning and domain generalization. We illustrate the suitability of PA by evaluating several models under different nature and magnitudes of shift, and proportion of affected observations. The results show that PA offers a reliable analysis of the vulnerabilities in learning algorithms across different shift conditions and provides higher discriminability than accuracy-based measures, while requiring no supervision.

---

[*]Equal contribution, authors listed in alphabetical order.
[†]Corresponding author.

# 1 Introduction

Real-world data analysis problems are often formulated as (potentially intractable) combinatorial optimization tasks, *e.g.*, inferring data clusterings, segmentations, embeddings, parameter estimation, etc. The stochastic nature of the input data and the computational complexity of the involved information processing require resorting to approximated, probabilistic estimates. However, such solutions often suffer from instability when generalizing to new observations that contain the same signal but systematically different noise perturbations, a phenomenon often referred to as *covariate shift* (Quiñonero-Candela et al., 2008). Covariate shift has gained increasing relevance, especially with the rise of deep learning and its astonishing improvements for a wide variety of predictive tasks. The focus of the research community is progressively "shifting" towards novel models that cover different levels of heterogeneity in the data and their structure. Consequently, new experimental settings have been implemented in order to test the robustness of machine learning models under nontrivial perturbations of the signal.

The focus of this work is on image classification tasks in two main categories of covariate shift:

- *adversarial (intentional) shift*: data is crafted ad-hoc by an adversary with vicious intentions, to hinder the output quality of the algorithm (Carlini & Wagner, 2017b; Biggio & Roli, 2018).

- *out-of-distribution (unintentional) shift*: data is subject to different initial conditions during its measurement (*e.g.*, lighting conditions, orientation, and so on)[1] (Koh et al., 2021; Wang et al., 2023a).

Under these settings, the usual standard robustness evaluation procedures consist of comparing the task performance of the model by computing a performance measure – *i.e.*, the accuracy or a derived alternative – under increasing levels of shift (Cinà et al., 2025; Koh et al., 2021). Therefore, such an evaluation is the same commonly used to test the generalization capabilities of models in the absence of measurable/accountable shift, that is, when data is subject to the randomness entailed by the sampling process only. In the following, we argue that the assessment of model robustness requires a paradigm shift in the way it is approached.

We first characterize the concept of robustness measure with two ideal properties that it should possess:

1. *non-increasing*: the measure should be non-increasing with respect to the response of the model against increasing levels of the shift power.

2. *shift-sensitive*: the measure should differentiate models *only* by their response against covariate shift.

In essence, Property 1 requires the measure not to increase the robustness score of a model if the shift in the data increases. In the best case, a model should be insensitive to shift, therefore scoring the same robustness as when it is absent. Property 2 requires, instead, that the measure reacts only to covariate shift and is not, for instance, sensitive to in-distribution sampling randomness and its related task performance.

Such properties serve as guiding principles for the evaluation process and are intentionally not formalized. This qualitative reasoning is due to the concept of robustness depending, case-by-case, on the definition of *shift power*. For example, in adversarial learning (Madry et al., 2018), the shift power usually coincides with the norm of the difference between a data observation and its perturbed counterpart. On the other hand, in domain generalization (Eulig et al., 2021), it usually encodes perceptual or semantic alterations that are specific to each dataset and that are hard or even impossible to characterize in a unified way. Furthermore, the shift power may also describe different aspects of the perturbation, such as the number of affected observations, which is often overlooked but equally relevant to time and performance efficiency. In both settings, a principled approach to measuring robustness is required and is currently missing. Another difficulty, related to Property 2, arises from the fact that it is not always possible to separate the covariate shift from the sampling noise, as this would require access to the often unknown data-generating process.

In general, accuracy-based measures decline with growing shift power, thereby satisfying Property 1. However, as accuracy results from thresholding over the predictions, it is unable to encapsulate any information

---

[1]In some cases, out-of-distribution is used to define a shift in the target set (Fang et al., 2022).

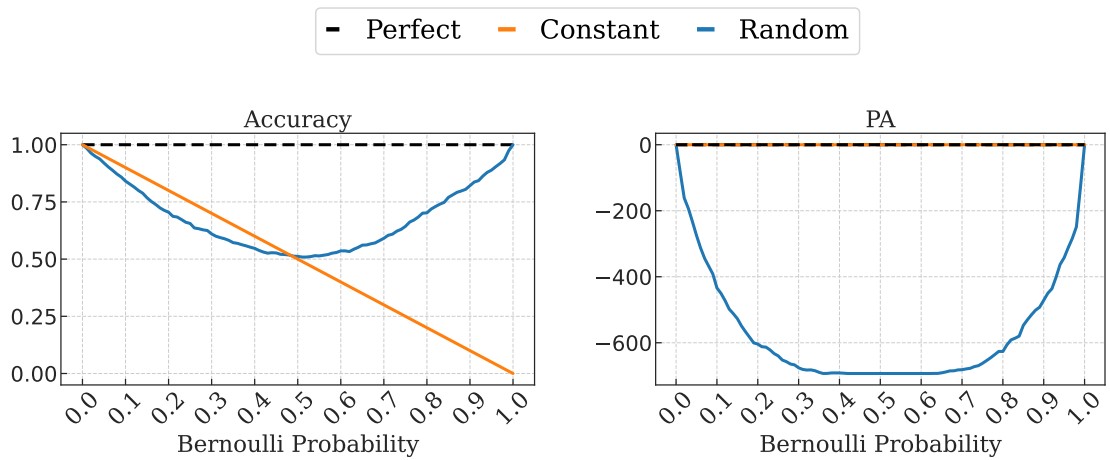

Figure 1: Comparison of accuracy and our proposed measure, Posterior Agreement (PA), to assess the robustness of three binary classifiers. We simulate a dataset of targets $D_Y = \{y_1, \ldots, y_n\}$, $|D_Y| = N$ where each $y_i \sim \mathcal{B}(p)$ is sampled from a Bernoulli distribution with $p = p(Y' = 1)$ (displayed on the $x$-axis). We assume a perfect classifier $f(y_i) = y_i$, a constant classifier $f(y_i) = 0$, and a random classifier whose outputs are a permutation of $D_Y$, so that the number of mismatched observations depends on $p$. Accuracy does not comply with the desired properties of a robustness measure and provides an inconsistent assessment, which is exclusively driven by task performance. Instead, PA detects the robustness of a constant classifier, discriminating it from the random, unrobust one.

regarding model confidence. In standard validation procedures, devising prediction-confidence-based measures increases their discriminative power, especially when comparing models with similar predictive capabilities but different confidence in the predictions. In the covariate shift setting, this distinction is decisive, not only for characterizing a model's robustness even in the presence of low shift power perturbations but also for providing a more consistent robustness assessment that does not heavily depend on the stochasticity of the dataset. Additionally, some accuracy measures involve a comparison with the target variables in the test set (*e.g.*, the attack success rate in adversarial learning), thus not complying with Property 2. For example, a *constant classifier* outputting the same prediction, regardless of the input, is robust by definition, since its output is independent of any shift in the data, thus covariate shift too. Similarly, a *perfect classifier* that always outputs the target's input is also robust. An accuracy measure would, however, discriminate the two models as differently robust since their performance would differ (*cf.* Figure 1, left).

In this work, we address these weaknesses and propose a new measure that better complies with the aforementioned properties, from the principled perspective of the Posterior Agreement (PA) framework (Buhmann, 2010; Buhmann et al., 2018a; Gronskiy, 2018). PA is an alternative model validation procedure rooted in information-theoretic thinking, in particular the rate-distortion theory (Cover & Thomas, 1999). In this setting, the learning algorithm is thought of as a lossy compression procedure with the aim of resolving the hypothesis class as precisely as possible, given the stochastic data source, a critical requirement for a measure aligned with the two proposed properties. The proposed PA measure provides a unique and unified framework for robustness assessment in the covariate shift setting, as it relies on a concept of robustness that stem from the consistency of the probabilistic response of the model under different data instances. PA is a confidence-based measure that does not depend on a model's classification performance, thus aligning with the foundational properties outlined before. As an illustrative example, Figure 1 depicts the difference between accuracy and PA in the robustness assessment of the constant/perfect classifiers described before. Since, the prediction confidence of a constant model is unaffected by the data, PA awards its robustness.

The original posterior agreement framework was devised to provide an epistemologically grounded methodology for finding robust solutions to optimization problems. In short, the method is given two dataset instances, used to fit two posterior distributions. The spread of such distributions is governed by a temperature pa-

rameter that trades off their informativeness and stability, in order to reach an *agreement* and reconcile their differences due to the sampling noise in the data. In this article, we focus on leveraging the PA framework for robustness evaluation. In particular, we assume that the shift between the two data realizations is covariate. By using the PA approach, we quantify the agreement between the related posteriors. The obtained measure estimates, thus, the robustness of a tested model as its capability to treat the shift as generalizable sampling noise. In such a case, the posteriors strongly agree, and our PA measure awards the model a good score.

Our work proceeds as follows. We first present the original PA framework and adapt it to covariate shift settings without modifying its fundamental principles. In particular, we formulate PA without any requirements of supervised information, making it a suitable measure for robustness evaluation in contexts where the supervision is very scarce or null (*e.g.*, medical applications, or autonomous driving). We then conduct a robustness analysis on two current settings of interest, namely adversarial learning in the evasion attack scenario and domain shift under targeted covariate shifts. In particular, we analyze the performance of several models in terms of PA under different natures and magnitudes of perturbation and proportion of affected observations. Not only does PA succeed in documenting the worsening of model performance in great detail, but it also detects an ongoing shift even when the number of targeted observations is low. Additionally, these findings were extended by leveraging PA for robust model selection with early stopping. In particular, experiments included the perturbation of data with spurious factor co-occurrences. The results show that PA-based selection succeeds at mitigating the spurious correlations existing between shifted environments.

In summary, *(i)* we propose the PA measure to assess model robustness in covariate shift scenarios and to motivate it theoretically, *(ii)* we conduct a robustness analysis on several settings, illustrating the findings emerging from an evaluation with PA, *(iii)* we summarize our findings and discuss a new notion of model robustness *in the PA sense* arising from the presented results.

## 2 Related Work

**Posterior Agreement**  Since its inception, the PA framework has been adapted to several different settings and is present in the literature under different variants, reflecting its evolution over time. The earliest version is called Approximate Set Coding (ASC, Buhmann, 2010; 2012), that interprets a model selection problem as an imaginary communication channel entailing a capacity optimization process. At the core of ASC lies the trade-off between informativeness and robustness of the selected solution. ASC works with *approximation sets*, *i.e.*, discrete subsets of hypotheses spanned by a cost function of the problem. Such sets are controlled by a tunable parameter determining an optimal trade-off between informativeness and stability of the solutions, discussed in more detail later. A variant of the framework goes under the name of SIMILARITY approach (Buhmann et al., 2018b), where approximation sets are tuned to maximize the ratio of solutions expected due to the similarity of the data instances. A recent version of the framework (Buhmann et al., 2018a) applies a relaxation of the ASC approach with posterior distributions that will be presented in this work. Here, approximation sets are replaced by Gibbs posteriors tunable through a temperature parameter. In the following, we will refer to this method as PA. In all its variants, PA has been applied to a variety of different selection and validation problems, including clustering (Buhmann, 2010), regression (Gorbach et al., 2017; Wegmayr & Buhmann, 2021), sorting (Busse et al., 2013), MAXCUT problems (Gronskiy & Buhmann, 2014; Bian et al., 2016), and assignment problems (Buhmann et al., 2014; 2017; 2018a).

**Adversarial Learning**  Adversarial robustness is a fundamental research area in machine learning due to the susceptibility of deep neural networks to small perturbations, leading to erroneous predictions (Szegedy et al., 2014; Goodfellow et al., 2015). A huge plethora of effective and efficient attacks such as FGSM (Goodfellow et al., 2015; Yuan et al., 2019), PGD (Kurakin et al., 2018; Madry et al., 2018), BIM (Kurakin et al., 2018), C&W (Carlini & Wagner, 2017b), DeepFool (Moosavi-Dezfooli et al., 2016), FMN (Pintor et al., 2021), and others (Carlini & Wagner, 2017a; Croce & Hein, 2019; Modas et al., 2019; Zheng et al., 2023), have been developed to evaluate the adversarial robustness of machine learning models. Adversarial examples are often measured in terms of their distance from the original input, using $\ell_p$ norms (Carlini & Wagner, 2017b). Their use enables a consistent and well-defined way to measure the magnitude of the perturbation and to compare different adversarial attacks (Carlini & Wagner, 2017b; Cinà et al., 2025). In this work, we also include analyses on the ratio of perturbed observations in a dataset. We believe that

this aspect can help in better understanding the behavior of models against attacks, as later shown in the experimental section.

Assessing model robustness proved to be necessary for developing more reliable machine learning systems and for ensuring their resilience *before* deployment. Typically, white-box models are employed to analyze a model's worst-case scenarios and avoid relying on *security by obscurity* (Carlini et al., 2019; Eisenhofer et al., 2021; Däubener et al., 2020). The assessment is commonly performed in the literature by scoring models with accuracy-based measures such as *attack success rate* and *adversarial accuracy*. While the definition of these two scores is not consolidated, in general, they estimate the model performance over a test set in terms of un/successful attacks. Additionally, Weng et al. (2018) approach robustness evaluation as a Lipschitz constant estimation problem. The method is, however, susceptible to gradient masking (Goodfellow, 2018), causing the measure to overestimate the size of the perturbation needed to fool a model. In addition, the bound is obtained by exploiting the ReLU property of neural networks, therefore narrowing its scope to such models. Wang et al. (2023b) uses, instead, the converging time to an adversarial sample for estimating the robustness of a model. This approach requires the estimation of a Jacobian with respect to the input, which may be infeasible in the case of large-sized data, and is applicable only to differentiable models.

**Domain Generalization**  Domain generalization is a field of paramount importance in machine learning due to both the limitation of acquiring a diverse and large enough set of training samples (Wolpert & Macready, 1997), and the nature of current methods to capture spurious correlations from training data that lead to catastrophic loss of performance in out-of-distribution settings (Wiles et al., 2022; Geirhos et al., 2020). Over the years, several benchmark datasets have been introduced specifically for domain generalization. The PACS dataset (Yu et al., 2022) represents shifts in visual styles between real photos and artistic media like paintings, cartoons, and sketches, reflecting variations in colour, texture, and abstractness. VLCS (Torralba & Efros, 2011) captures object-centric and scene-centric distribution shifts, with variations in object variability, scene context, and annotation styles across other previously established datasets. The WILDS benchmark (Koh et al., 2021) embodies real-world shifts, covering temporal and geographical changes, as well as differences in image resolution and quality, which are typical in practical applications. Each benchmark encapsulates unique distribution shifts, providing diverse challenges for evaluating domain generalization methods. More recently, the DiagViB-6 dataset (Eulig et al., 2021) has also been suggested for the systematic analysis of data distribution shifts over multiple generative factors, covering hue, position, lightness, scale, and texture, and allowing for the accumulation of multiple shifts. While several new benchmarks have been proposed to address growing requirements for both more realistic and challenging evaluation settings, accuracy on the unseen domains remains the fundamental measure for estimating model robustness.

## 3  Methodology

In this section, we devise a tractable version of the posterior agreement for the validation of classification models in covariate shift settings. We first introduce the original PA framework and then adapt it in the context of a classification task. Similar to Buhmann (2010), we take our inspiration from simulated annealing and its deterministic variants (Rose et al., 1992; Buhmann & Kühnel, 1993; Rose, 1998). Our notation follows Buhmann et al. (2018a).

### 3.1  Background

In the original posterior agreement framework, we are given two datasets $X'$, $X'' \in \mathcal{X}$, derived from the same experimental settings, thus, with identical signal, but different noise realizations. In classic optimization settings, both data instances can be used separately to train a model, *i.e.*, to select a solution $c$ from a hypothesis set $\mathcal{C}$. Such solutions are, in general, distinct and often explain not only the signal but also part of the noise. To counter this overfitting, the PA approach (Figure 2) focuses on modelling two posteriors $p(c \mid X')$, $p(c \mid X'')$. In particular, PA mitigates the effect of the sampling noise by modulating the dispersion of the distributions. Large dispersion of a flat posterior can explain many different data samples and, therefore, they are less informative. On the other hand, small dispersion of a peaked posterior is highly informative but does not generalize well over sampling noise, and is, thus, less stable. PA aims at balancing

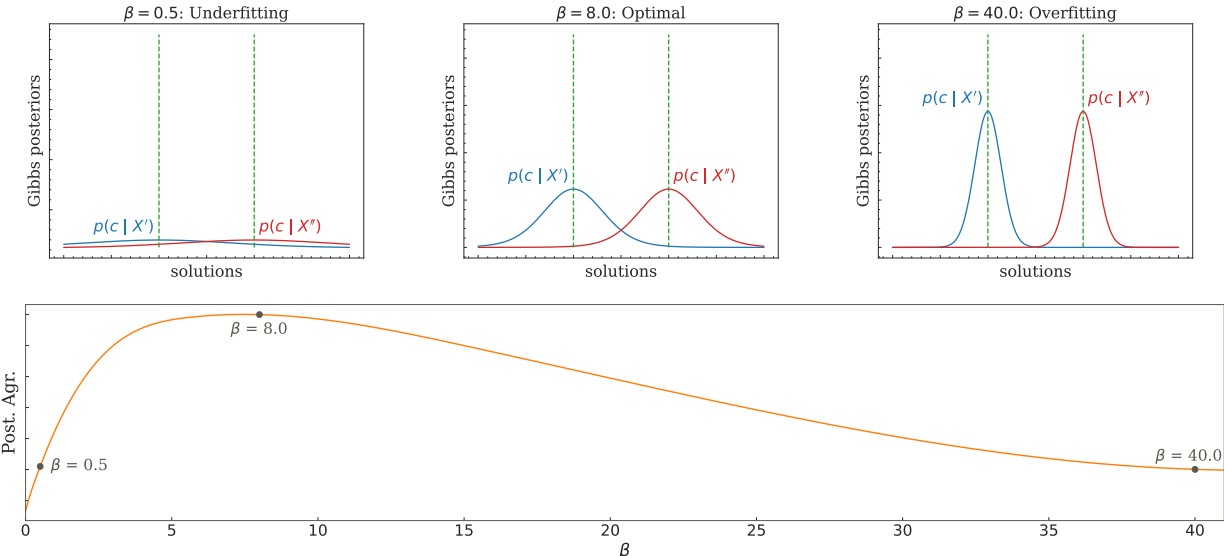

Figure 2: The PA framework, applied in the estimation of a real-valued parameter (*e.g.*, a distribution's mean). Two posteriors $p(c \mid X')$, $p(c \mid X'')$, $c \in \mathbb{R}$ are fit to two different data realizations. The optimization over the inverse temperature parameter $\beta$ is required to make the two posteriors insensitive to sampling noise. When $\beta$ is too low (top left), the posteriors tend to be uniform and uninformative, *i.e.*, they underfit. When $\beta$ is too high (top right), the posteriors are too peaked and therefore sensitive to noise perturbation, *i.e.*, they overfit. An optimal $\beta$ (top center) maximizes the posterior agreement (bottom), ensuring informativeness and stability that can be then used for model selection or, in our case, for robustness assessment. Illustration based on Buhmann et al. (2018a).

these two aspects, informativeness and stability, to achieve an *optimal agreement* between the posteriors that explains the selected solutions. The obtained distributions can then be combined and used to sample a more robust solution that better generalizes over the two noisy data realizations.

Traditionally, posteriors have been designed by calculating or approximating the respective Gibbs distribution defined by the score function $\mathcal{S}(c, X)$ used in the optimization problem. In PA, both Gibbs distributions are parameterized by a shared inverse computational temperature $\beta$ controlling their dispersion and their agreement. An optimal control of $\beta$ renders the posteriors informative and insensitive to sampling noise perturbation (*cf.* Figure 2, top).

In our case, $X'$, $X''$ represent a test set and its shifted instance, while the Gibbs posteriors stem from the model predictions of a trained model. The optimal agreement can then be quantified and used as a measure to validate the robustness against covariate shift. A robust model will generate similar posteriors, thus a high agreement. In a brittle model with lack of robustness, the posteriors will differ substantially.

## 3.2 Setting

Let $D_X = (x_i)_{i=1}^N$ be a dataset[2] of i.i.d. measurements (observations) drawn from a random variable $X$ with support $\mathcal{X}$. A $K$-class *classifier* can be defined as the composition of two functions:

- a function $F : \mathcal{X} \to \mathbb{R}^K$, mapping observations to a tuple of *class conditional score functions*, *i.e.*, $x \mapsto F(x) = (F_1(x), \ldots, F_K(x))$. These functions are parametrized by a set of parameters $\theta$, which is fixed, as we are conducting an evaluation. The score term $F_k(x) \in \mathbb{R}$ quantifies the membership degree of observation $x$ to class $k$ (the higher, the more probable). $F_k(x)$ can be, for example, the logit of a multi-class prediction model (*e.g.*, a neural network).

---

[2]Note that we do not need to specify the targets of $D_X$ for assessing the robustness of a model.

- a *decision rule* $f : \mathbb{R}^K \to \{1, \dots, K\}$, applied to choose the class for each observation, based on the class conditional score functions. Usually, $f$ is the *Maximum A Posteriori (MAP)* rule:

$$f(F(x)) = \arg\max_j F_j(x) \tag{1}$$

A classifier is defined as $c = f \circ F$.

Note that, since $F$ is fixed, the hypothesis set is spanned by all possible decision rules over $\mathcal{X}$. Restricting the process to $D_X$, gives rise to a finite hypothesis set $\mathcal{C}$, containing all possible mappings from $c : D_X \to \{1, \dots K\}$, induced by a specific classifier $c \in \mathcal{C}$ [3]. Therefore, $|\mathcal{C}| = K^N$.

Each classifier is associated with a *score* evaluating its *confidence* in explaining the data:

$$\mathcal{S}(c, X; \theta) = \sum_{i=1}^{N} F_{c(x_i)}(x_i; \theta). \tag{2}$$

Such a score can be used to define a posterior $p(c \mid X)$ over the hypothesis set $\mathcal{C}$. In particular, we aim to find a probability distribution $p$ such that $\mathcal{S}(c_1, X; \theta) > \mathcal{S}(c_2, X; \theta) \iff p(c_1 \mid X) > p(c_2 \mid X), \forall c_1, c_2 \in \mathcal{C}$, a property that we will rewrite as $\phi_{\mathcal{C}}(p)$. As many distributions fulfill this requirement, we narrow the possibilities by applying the *Maximum Entropy Principle (MEP)* (Jaynes, 1957), which states to search for the maximally uninformative (*i.e.*, entropy-maximizing) distribution which best encodes the observed data.

The following optimization objective

$$\underset{p(c \mid X)}{\text{maximize}} \quad H[p] \tag{3a}$$

$$\text{subject to} \quad \phi_{\mathcal{C}}(p), \tag{3b}$$

$$p(c \mid X) \geq 0, \tag{3c}$$

$$\sum_{c \in \mathcal{C}} p(c \mid X) = 1, \tag{3d}$$

$$\mathbb{E}_{c \mid X}[\mathcal{S}(c, X)] = \mu \tag{3e}$$

gives the required solution. Here, $\mu$ is a hyperparameter ensuring that the expectation is finite. We set the Lagrangian without the inequality constraints

$$\mathcal{L}(p, \alpha, \beta) = H[p] \tag{4}$$

$$+ \alpha \left( 1 - \sum_{c \in \mathcal{C}} p(c \mid X) \right) \tag{5}$$

$$+ \beta(\mathbb{E}[\mathcal{S}(c, X) - \mu]), \tag{6}$$

and verify that they hold for the found solution. The derivative with respect to $p(c)$ is

$$\frac{\partial \mathcal{L}}{\partial p(c \mid X)} = -1 - \log p(c \mid X) - \alpha + \beta \mathcal{S}(c, X). \tag{7}$$

Equating it to 0 and solving for $p(c)$ gives

$$p(c \mid X) = \frac{\exp(\beta \mathcal{S}(c, X))}{\exp(1 + \alpha)}. \tag{8}$$

Setting $\exp(1 + \alpha) = \sum_{c \in \mathcal{C}} \exp(\beta \mathcal{S}(c, X))$ and $\beta \geq 0$ ensures that $p(c \mid X)$ is a Gibbs distribution that satisfies the constraints 3b and 3c with the inverse temperature parameter $\beta$. The exact value of $\beta$ depends on $\mu$, and can be found by enforcing the posterior agreement principle, discussed in the next section.

The summation over the hypothesis set can pose a drawback in the computation of the posterior. The following result provides an efficient factorization for $p(c \mid X)$.

---

[3]The finiteness of $\mathcal{C}$ is a mathematical expedient to efficiently deal with the subsequent derivations. By restricting to $D_X$, $c$ represents an index mapping on a finite set, which is less discriminative than $c = f \circ F$ defined above. *cf.* also Section 5.

**Theorem 1.**

$$p(c \mid X) = \prod_{i=1}^{N} p_i(c(x_i) \mid X),\tag{9}$$

*where*

$$p_i(k \mid X) = \frac{\exp(\beta F_k(x_i))}{\sum_{j=1}^{K} \exp(\beta F_j(x_i))}\tag{10}$$

*is the probability that $x_i$ is assigned to class $k$.*

*Proof. cf.* Appendix A.1 □

### 3.3 Posterior Agreement

We are given two i.i.d. datasets $X'$, $X''$ with $|X'| = |X''| = N$ and for $i = \{1, \ldots, N\}$ $x_i'$ and $x_i''$ are two realizations sampled from the same ideal (*i.e.*, noiseless) process $x_i^{(0)}$. Given a classification model, we can estimate its agreement in the prediction results through the *expected posterior agreement kernel*

$$k = \mathbb{E}_{X', X''} \left[ \log \left( \sum_{c \in \mathcal{C}} \frac{p(c \mid X')p(c \mid X'')}{p(c)} \right) \right],\tag{11}$$

where the expectation is computed over the random variables of $X'$ and $X''$. A notable difference with the previous versions of the kernel is the addition of the term $p(c)$. It is a prior over $\mathcal{C}$, used to account for the model complexity. In the following, we will work under the assumption that we do not have access to any prior information, that is $p(c) = 1/|\mathcal{C}|$, $\forall c \in \mathcal{C}$. Often, the computation of the expected value over $X'$ and $X''$ is infeasible, therefore the *empirical posterior agreement kernel* is adopted as

$$k(X', X'') = \log \left( \sum_{c \in \mathcal{C}} \frac{p(c \mid X')p(c \mid X'')}{p(c)} \right),\tag{12}$$

which estimates the overlap between the Gibbs posteriors defined as explained above.

The computation depends on the inverse temperature hyperparameter $\beta \in \mathbb{R}_{\geq 0}$. Similar to the MEP, we search for the $\beta$ that maximizes the overlap between the posteriors. The *Posterior Agreement (PA)* measure is then computed as

$$\text{PA}(X', X'') = \underset{\beta}{\text{maximize}} \quad \frac{1}{N} k(X', X'').\tag{13a}$$

$$\text{subject to} \quad \beta \geq 0 \tag{13b}$$

$1/N$ is a scaling correction factor since PA increases proportionally with the size of the dataset.

Below, we propose and discuss an operative formula to compute the empirical posterior agreement kernel.

**Theorem 2.** *With no prior information available, the empirical posterior agreement kernel $k(X', X'')$, can be rewritten as:*

$$k(X', X'') = \log \left( |\mathcal{C}| \prod_{i=1}^{N} \sum_{j=1}^{K} p_i(j \mid X') p_i(j \mid X'') \right).\tag{14}$$

*Proof. cf.* Appendix A.2. □

By optimizing Program 13 with this kernel, we can see the effect of $\beta$ in determining the optimal overlap between posteriors. An increment in $\beta$ corresponds in peaking the distributions $p_i(\cdot \mid X)$, $X = X', X''$ toward their MAP, awarding *matching distributions* that share the same MAP. On the contrary, decreasing $\beta$ flattens the distributions, and mitigates penalizations due to *mismatching distributions* with different

MAPs. Therefore, the optimization operates a tradeoff between these opposite sub-objectives, searching for the best solution that explains the agreement between the two data sources.

We conclude by characterizing the posterior agreement measure as computed with the kernel of Theorem 2.

**Theorem 3.** *Under no prior information available, the PA measure has the following properties:*

1. *Boundedness:* $0 \leq PA(X', X'') \leq \log K$.

2. *Symmetry:* $PA(X', X'') = PA(X'', X')$.

3. *Concavity:* $PA(X', X'')$ *is a concave function in* $\beta < +\infty$.

*Proof. cf.* Appendix A.3. □

## 4 Experimental Results

In this section, we present a comprehensive analysis of the empirical results. In particular, we study two covariate shift scenarios, adversarial learning and domain generalization, and set up a comparison of the scores obtained by different models, using PA and accuracy-based measures. Our main purpose is to highlight the differences between performance- and robustness-based evaluation criteria. Therefore, we compare several defences and learning techniques with different capabilities, to encompass, as much as possible, the variety of cases that can arise during an evaluation process. For further technical information, the reader is referred to our code implementation[4].

For measuring model robustness, we use the PA measure with no prior information. In particular, we use the original test set as $X'$ and its perturbed version as $X''$. For a better visualization of the results, we adopt a logarithmic scale, and we omit the $|\mathcal{C}|$ and the $1/N$ constants in PA to increase its dynamic range, therefore PA $\in [-N \log(K), 0]$.

### 4.1 Adversarial Robustness

For the adversarial robustness scenarios, we carry out our experiments with the CIFAR-10 (Krizhevsky et al., 2009) and the ImageNet (Deng et al., 2009) datasets, widely adopted in the machine learning security literature as benchmarks for robustness evaluation. The CIFAR-10 dataset contains $60\,000$ colour images of $32 \times 32$ pixels equally distributed in 10 classes. The analyzed models are trained on the training set ($50\,000$ images), and the PA evaluation is performed on the test set ($10\,000$ images). The ImageNet dataset contains circa 1.5M colour images processed at $256 \times 256$ pixels, unequally distributed in 1000 classes. In this case, the evaluation is conducted on the validation set, as is usually done in the literature. A random subset of $10\,000$ images is selected, so that finite-sample effects and PA values are comparable to those in CIFAR-10 experiments. In both cases, we perturb the test set using several evasion attack methods. We start from two simple attacks, Projected Gradient Descent (PGD, Madry et al., 2018) and Fast Minimum Norm (FMN, Pintor et al., 2021), and we proceed to more efficient benchmark attacks such as APGD and FAB from the AutoAttack library (Croce & Hein, 2020b). The attack power is specified in terms of the $\ell_\infty$ norm and is set in advance ($\ell_\infty = \ell/225$, $\ell \in [2, 4, 8, 16, 32]$) for all attacks except for FMN, where the attack norm is automatically searched for each observation. All attacks are run on the evaluation set to produce perturbed versions for subsequent PA computation. PGD and FMN attacks are run for 1000 steps, while the number of steps in AutoAttack is determined automatically. In the following, we discuss the results for PGD and FMN in CIFAR-10, while we include the remaining AutoAttack experiments in Appendix F.

We consider several models with different robustness properties. In particular, we score an undefended WideResNet-28-10 (Zagoruyko & Komodakis, 2016) and five defended models: a ResNet18 (He et al., 2016a) defended by Addepalli et al. (2022), a ResNet50 (He et al., 2016a) defended by Engstrom et al. (2019), a WideResNet-28-10 defended by Wang et al. (2023c), a PreActResNet18 (He et al., 2016b) defended according

---

[4]PA: https://github.com/viictorjimenezzz/pa-metric
Experiments: https://github.com/viictorjimenezzz/pa-covariate-shift.

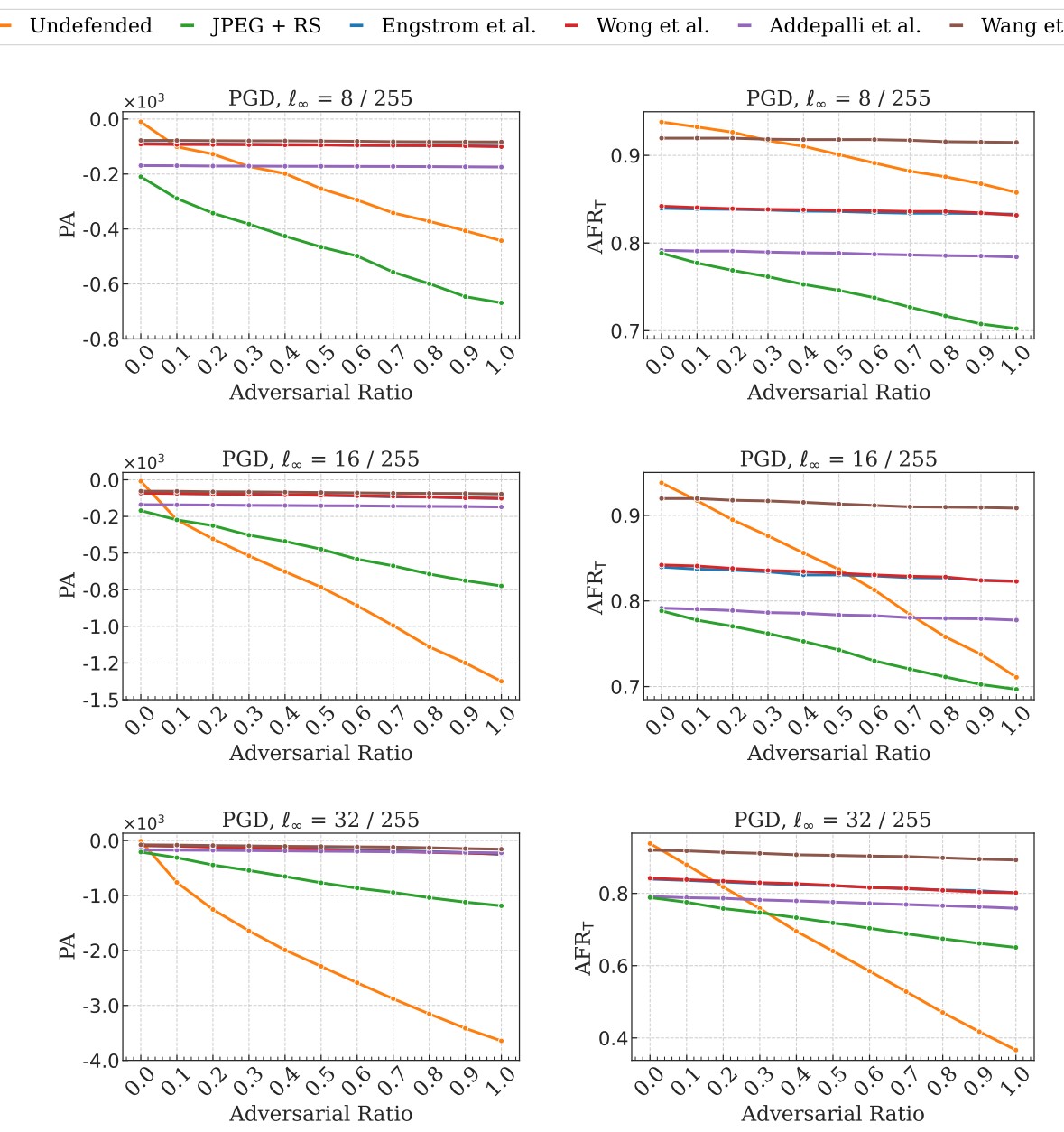

Figure 3: PA (left) and $\text{AFR}_T$ (right) scores against increasing AR and $\ell_\infty$, for the PGD attack. The tendencies are similar with the only exception of the undefended model, which is overperforming according to $\text{AFR}_T$.

to Wong et al. (2020) and a 3-layer CNN (LeCun et al., 2015) defended by a JPEG pre-processing Das et al. (2017) and a Reverse Sigmoid post-processing Lee et al. (2018). All models except for the latter are implemented with the RobustBench library (Croce et al., 2021).

We perform a comparison of PA with an accuracy-based measure, the Attack Failure Rate ($\text{AFR}_T = 1 - \text{ASR}_T$) and evaluate the performance of the models under investigation to verify the effectiveness of the PGD and FMN attacks. The evaluation is performed over ten different adversarial ratios (ARs) of attacked examples in the test set, ordered by increasing attack power. In particular, a dataset with $AR = p$ contains the attacked observations under the $10p$-decile, ordered by $\ell_\infty$, and the remaining unattacked ones. $AR = 1$

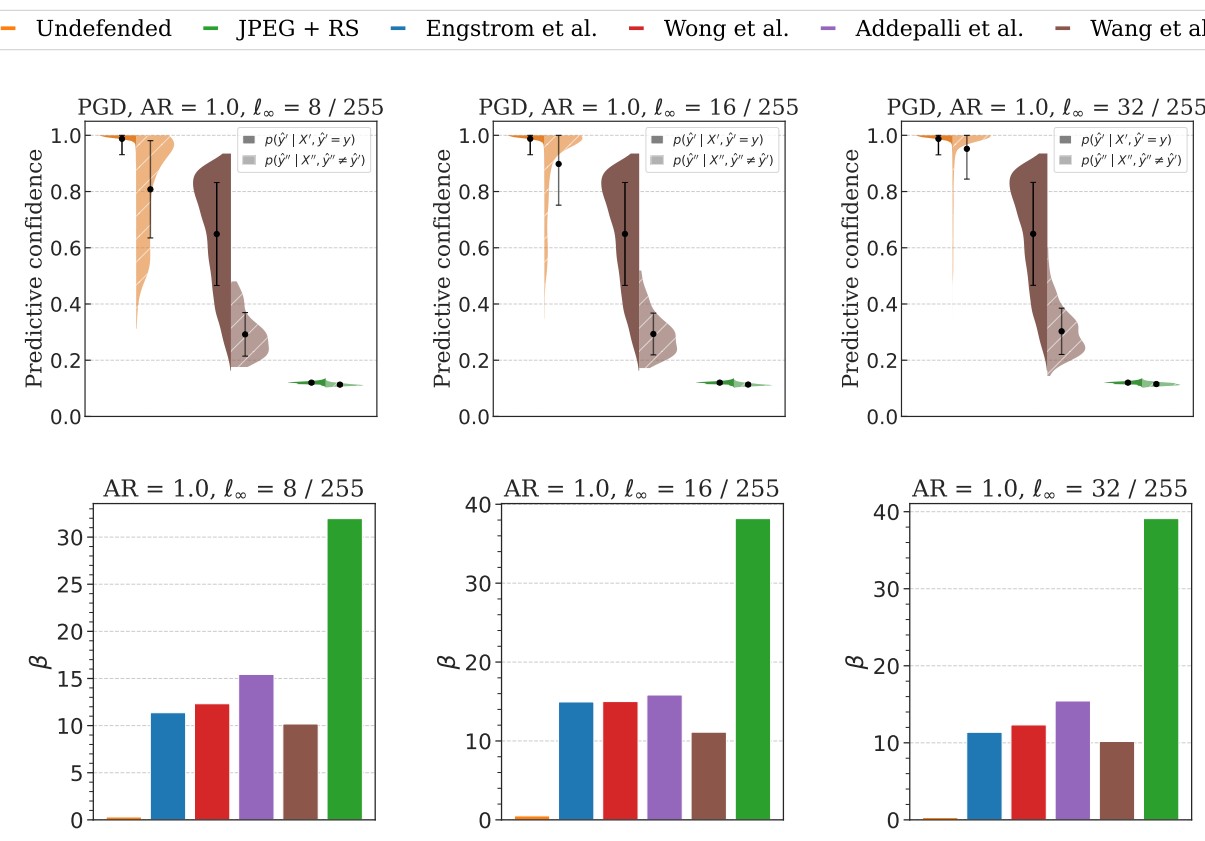

Figure 4: (top) Distribution of the predictive confidence for three example models. The robust one (Wang et al., 2023c) is significantly decreasing its confidence, while the weak ones (undefended and JPEG + RS) are not. (bottom) Final $\beta$ values for $AR = 1$. Anomalous $\beta$ values identify the weak models.

corresponds to an entirely attacked dataset. The $\beta$ parameter is searched with an Adam (Kingma & Ba, 2015) optimization procedure, run for 500 epochs. We attested a relatively, fast optimization of the $\beta$ parameter, usually in the order of tens of minutes in single GPU even for large dataset (*i.e.*, ImageNet).

**PGD Attack** As illustrated in Figure 3 (left), PA consistently discriminates the undefended model, which significantly decreases its performance with increasing adversarial ratio and attack power. As expected, the rate at which the performance decreases is faster the more powerful the attack is. The JPEG + RS model also displays poor robustness scores, compared to the other models, especially for low-norm attacks, which is consistent with the JPEG compression defence employed.

The fundamental difference between PA and $\text{AFR}_T$ arises in the scoring of the undefended model, which overperforms according to $\text{AFR}_T$, especially for low $\ell_\infty$. To understand the causes of such a difference, in Figure 4, we display the distribution in prediction confidence of three example models. In particular, we compare the correctly predicted observations before the attack ($p(\hat{y}' \mid X', \hat{y}' = y)$, full colour) with the successfully attacked ones ($p(\hat{y}'' \mid X'', \hat{y}'' \neq \hat{y}')$, streaked colour). We look at two weak models (the undefended and JPEG + RS) and the most robust one (Wang et al., 2023c), in the case of $AR = 1$ (*cf.* Appendix B for an analysis of all models). Standard errors (black bars) are also included. Additionally, we include the values of the optimized $\beta$ parameters after convergence for all the models. We analyze the behaviour of the three PGD plots:

- The undefended model outputs, on average, high-confidence predictions. The standard errors overlap, hinting that the model cannot detect that an attack has happened, therefore not lowering enough

its confidence. $\text{AFR}_T$ scores are high overall. The maximization of PA entails flat posteriors, to mitigate the effect of the mismatches between the pre- and post-attack distributions. Therefore, $\beta$ converges to a low value. Consequently, we obtain low PA scores overall.

- The JPEG + RS model provides low confidence predictions, due to the reverse sigmoid defense used, and does not reduce its predictive confidence in the presence of misleading adversarial examples. As a result, PA is maximized when posterior distributions are highly informative, to peak the matches (*i.e.*, $\beta$ converges to a high value). However, mismatching predictions induce a penalty on the robustness score that drives PA to a low value, overall. $\text{AFR}_T$ scores are low, as well.

- Wang et al. (2023c) is sensible to the attack and significantly lowers its overall confidence, as indicated by non-overlapping standard errors. Both $\text{AFR}_T$ and PA are high, overall.

In conclusion, while both measures report similar trends and rankings, AFR tends to overestimate the robustness of the undefended model, especially when a few number of samples have been attacked, while PA provides a more consistent assessment, unbiased by the nature of the experiment (*e.g.*, $\ell_\infty$ and $AR$).

Additionally, note that for $AR = 0$, PA scores do not converge to the theoretical result of 0. In the absence of shift, the optimal $\beta$ tends toward infinity to peak all coinciding distributions over their MAPs (*cf.* proof of Theorem 3, Property 2). However, the Adam optimizer progressively scales down the learning rate, dampening, in turn, the convergence rate. The obtained PA scores then reflect the "peakedness" of the initial model predictions, that is, they provide information about the predictive confidence of each model before the attack, with the most confident models scoring better. The ranking of PA coincides with that of $\text{AFR}_T$, hinting that in the absence of shift, the tested models yield higher confidence in positive predictions than in negative ones[5]. This additional insight is obtained thanks to the use of an adaptive optimization algorithm such as Adam, advocating for its use in the optimization of $\beta$.

**FMN Attack** FMN does not require an $\ell_\infty$ limit, and automatically finds, for each observation, the minimum distance needed to evade the classification. We found that for $AR \gtrsim 0.5$, the observations are perturbed far over the shift power against which all models are robustified. We, therefore, focus on the cases with $AR \leq 0.5$, and report, for completeness, the extended results in Appendix B. In Figure 5, the PA and $\text{AFR}_T$ scores are presented. Again, the undefended model robustness is overestimated by $\text{AFR}_T$. For some $AR$s, the JPEG + RS model performs better than the other models overall, showing that the JPEG compression defensive mechanism effectively filters out the perturbations and reduces the attack transferability performance of FMN. In Figure 6 (right), we can note the increment in the corresponding $\beta$, indicating that the two distributions contain more MAP matches. Wang et al. (2023c) results instead in being the weakest model and, therefore, less suitable against this attack. Accordingly, its $\beta$ is low. For completeness, in Appendix B, we include the predictive confidence distributions of all models for $AR \in \{0.5, 1\}$.

**Autoattack** In Appendix F, we present experiments conducted on CIFAR-10 and ImageNet with the Autoattack library (Croce & Hein, 2020b). Specifically, we report robustness scores for APGD-CE/DLR/DLR$^\top$, and FAB attacks on CIFAR-10. The results exhibit trends consistent with those discussed above: PA provides a reliable assessment under increasing shift power, whereas $\text{AFR}_T$ yields inconsistent rankings and fails to discriminate effectively between models. Below, we shortly discuss the obtained results. The reader is referred to the appendix for a more detailed discussion.

In the CIFAR-10 case, Autoattack is much more effective than the vanilla PGD attack discussed above. Evaluating a wider range of epsilon values reveals the full spectrum of $\text{AFR}_T$ scores and confirms the superior discriminability of PA even when all defenses fail. For example, PA ranks Addepalli et al. (2022) and Wang et al. (2023c) as the most robust defenses against APGD attacks across all $\ell_\infty$ and AR values, whereas $\text{AFR}_T$ only reaches this conclusion in the high shift power regime.

---

[5]However, this is not always necessarily true. We want to remark that task performance and predictive confidence are two distinct aspects (*cf.* Appendix C).

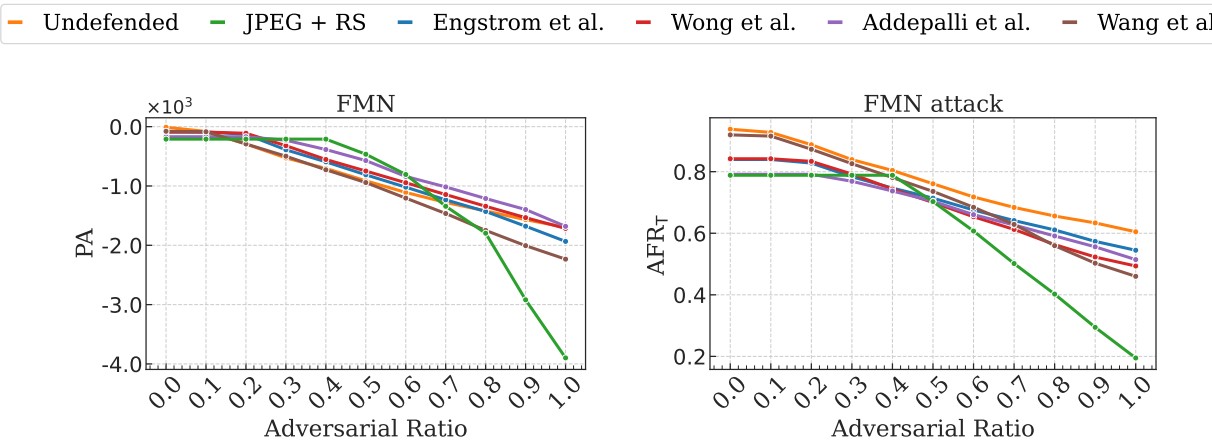

Figure 5: PA (left) and $\text{AFR}_T$ (right) scores against increasing AR and $\ell_\infty$, for the FMN attack. Again, the undefended model robustness is overestimated, according to $\text{AFR}_T$. For $AR \in [0.3, 0.6]$ JPEG + RS model is more robust than the others, with a similar trend in performance.

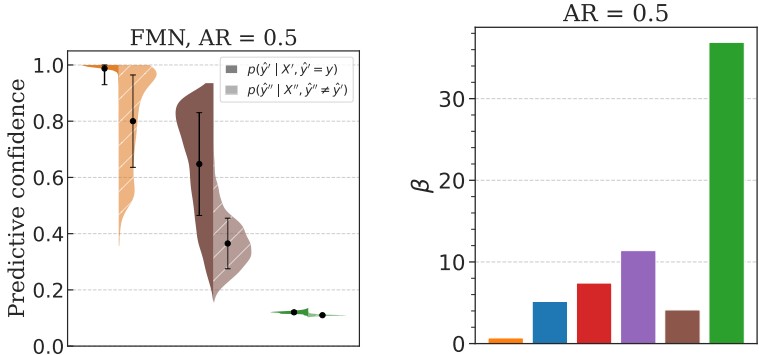

Figure 6: Predictive confidence distributions (left) and $\beta$ plots (right) for the FMN attack. The results are related to $AR = 0.5$ not to include perturbations against which the models are not robust. The models perform similarly to the PGD case, with a less marked difference in the standard error for Wang et al. (2023c) model.

In the ImageNet case, we observe that PA and $\text{AFR}_T$ yield the same defense ranking. This is due to the high success of the attack, which makes performance degradation drive the PA score. Still, PA demonstrates greater consistency, ranking models correctly when $AR > 0$, whereas $\text{AFR}_T$ requires $AR \gtrsim 0.8$.

## 4.2 Domain Generalization

Following the analysis on adversarial robustness, we perform a series of experiments on several domain generalization settings to assess both model discriminability and model selection when comparing PA to accuracy-based measures. Again, $\text{AFR}_T$ is taken as the comparison reference. In addition, Adam is run for 1000 epochs to search for the $\beta$ parameter.

In this scenario, we conduct our experiments through a modified version of the DiagViB-6 dataset (Eulig et al., 2021) that comprises distorted and upsampled coloured images of size $128 \times 128$ from the MNIST dataset (LeCun, 1998). Each image is defined by five controllable factors that induce changes in texture, hue, lightness, position, and scale. A *domain* corresponds to a specific configuration of these factors.

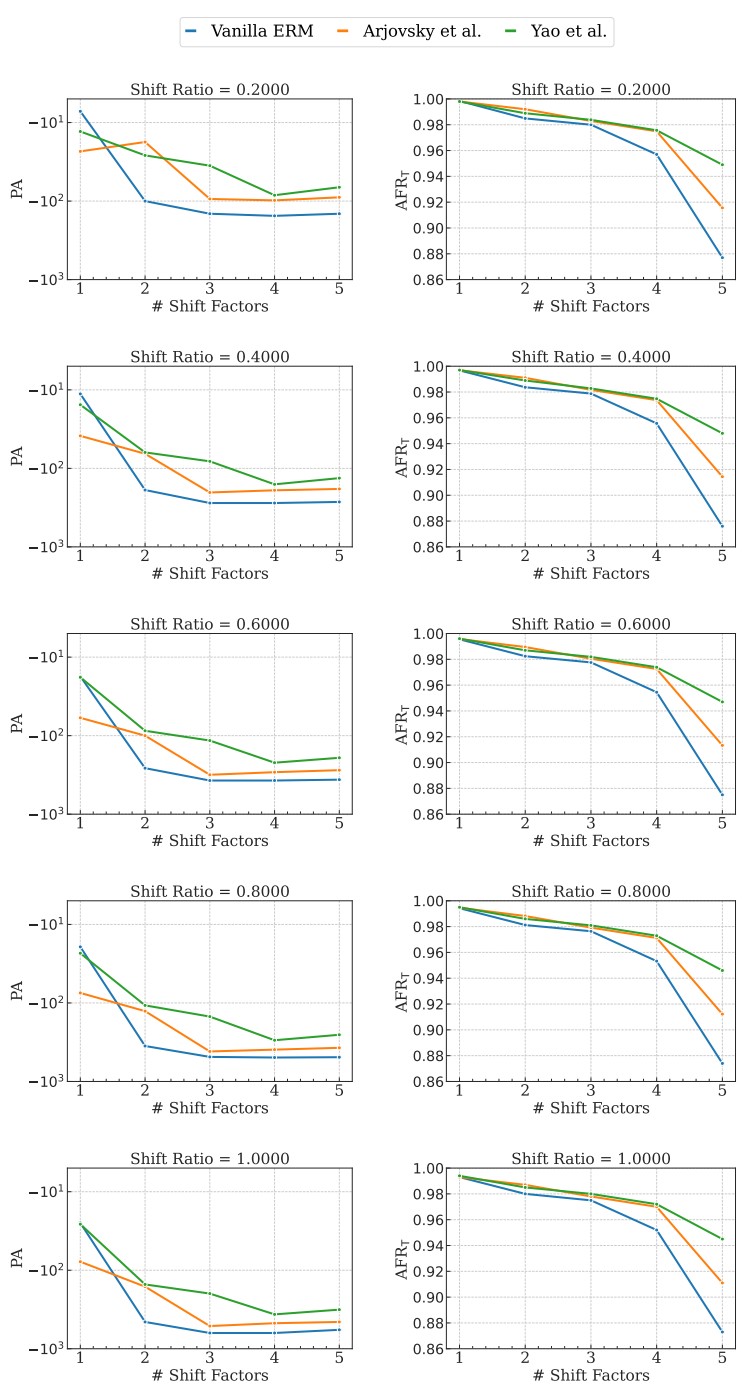

Figure 7: Effect of domain shift in the PA performance on a weak and robust model, over five distinct levels of the shift ratio. Each plot depicts the response of both models to cumulative levels of distribution shift, from one to five shifted factors. At each shift ratio, PA is able to differentiate between the two models.

For training, we define two unique original domains, $\{e_0^1, e_0^2\}$, under a pair of original instantiations of each image factor, where all the factors of $e_0^1$ are different from $e_0^2$. We induce a second pair of domains for the validation data, following precisely the factor configuration of the training domains, but with disjoint set of

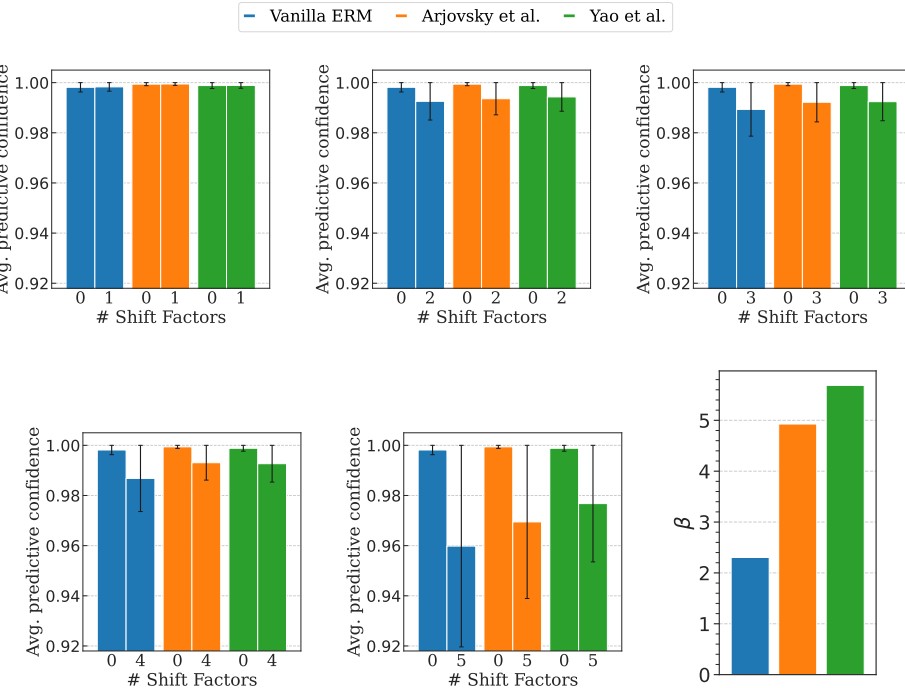

Figure 8: Average predictive confidence for three models—(blue) Vanilla ERM, (orange) IRM, and (green) LISA, under a $SF = 1$. Each panel shows results at a different number of shift factors ($\#SF$), illustrating how the models' predictive confidence varies as the distribution shift increases. The left column remains fixed, and the right column changes across $\#SF = 1, 2, 3, 4, 5$. The final bar plot on the right depicts the $\beta$ for 5 shift factors. We see that higher model confidence leads to an adjusted higher $\beta$.

samples. Finally, a set of six additional domains, $\{e_0, e_1, \ldots, e_5\}$, is generated. The factors of domain $e_0$ are identical $e_0^1$, while each $e_i (i = 1, \ldots, 5)$ is obtainined by modifying exacty $i$ of the five factors relative to $e_0^1$.

Our final dataset comprises two sets of 40 000 images for training, two sets of 20 000 images for validation, and six sets of 10 000 images for testing. Note that the original MNIST images used for generating the datasets do not overlap. For more details on how the dataset was adjusted to our setting, refer to Appendix E.

For the experiments, we consider a ResNet50 model trained with three different algorithms: vanilla Empirical Risk Minimization (ERM, Vapnik, 1991), Invariant Risk Minimization (IRM, Arjovsky et al., 2019) and Learning Invariant Predictors with Selective Augmentation (LISA, Yao et al., 2022). These methods were chosen as representative examples of two distinct approaches to domain generalization: IRM operates by enforcing invariance at the level of latent representations, while LISA augments the data itself to promote invariance in the input space. By including both, we evaluate robustness across approaches that intervene at different stages of the learning process.

**Model discriminability** In Figure 7, we showcase the PA evaluation results over cumulative distribution shifts. Each plot presents the variation of PA over five cumulative implementations of the shift factors ($SF$), in five distinct test sets corresponding to increasing amounts of shift ratio ($SR$). In particular, we subject a subset of the test set to the related cumulative distribution shift to an SR corresponding to 20%, 40%, 60%, 80%, and 100% of the original dataset. Similarly to $AR$, $SR = 1$ entails a change of all the samples in the dataset.

As depicted, and already well established in Koh et al. (2021), the overall trend of robustness starts with ERM as the weakest model, followed by IRM, and then by LISA as the most robust. However, PA is also able to identify more nuanced robustness behaviors. In particular, when $SF = 1$, ERM achieves a higher

PA score because it relies on domain-specific features already encountered during training, even when the number of mismatching predictions is higher. Note that PA is the only measure that was able to discriminate ERM. Conversely, the domain invariance imposed by IRM slightly hinders PA under these mild shifts, even though its overall AFR remains comparable or better.

This illustrates that PA and $\text{AFR}_T$ provide complementary insights in domain generalization: AFR reflects performance differences across shifts, while PA emphasizes the stability of predictive distributions under factor perturbations. For example, PA stabilizes after certain shifts because the model has not learned to represent additional factors (*e.g.*, after hue and brightness), whereas AFR continues to decrease as performance drops. At the same time, PA can capture fine-grained effects that $\text{AFR}_T$ may overlook. In particular, the source-versus-target domain difference highlights how PA reveals model behavior when domain shifts partially overlap with training distributions, while $\text{AFR}_T$ may mask these nuances. Consequently, IRM's robust design trades off some PA in low-shift regimes but does not sacrifice performance, distinguishing its behavior from both ERM and LISA in a way that PA can clearly reveal. Taken together, both measures are valuable for a comprehensive evaluation of robustness.

As shown in Figure 8, predictive confidence varies across models under distribution shifts. ERM starts with high confidence but declines as the number of shift factors increases, while IRM remains stable, and LISA consistently exhibits the highest confidence, which suggests superior robustness. Higher confidence models require a greater $\beta$ for posterior alignment, as seen in the rightmost plot. Conversely, lower confidence models benefit from reduced $\beta$ to flatten distributions and minimize mismatches. This effect is most pronounced with five shift factors, where LISA maintains strong confidence while ERM struggles. This behavior aligns with adversarial settings, where models with lower confidence require a higher $\beta$ to align their distributions. However, this is not strictly a matter of distribution alignment but rather *mutual information* maximization: such models only push $\beta$ to higher values if their predictions agree. PA demonstrates superior performance because it effectively navigates the trade-off between model performance and informativeness. In particular, models with lower confiPA effectively captures these dynamics, and through optimal selection of $\beta$ can enhance robustness under shifting distributions.

We also attempted to better understand the impact of controlled domain shifts and their impact on robustness assessment through PA (*cf.* Appendix D for more details). A phenomenon observed in this work is the decrease in classification performance as the number of shift factors increases, while the PA measure remains comparable or slightly improves. Specifically, at $SF = 5$, the feature space differs significantly (as shown in Figure 11) from those at $SF \leq 4$, leading to lower accuracy yet roughly similar (or slightly improved) robustness. This observation highlights that in more complex settings, where the effects of the covariate shift are directly controlled (as in the case of DiagViB-6), the robustness of the model does not stem solely from how out-of-distribution the samples are. Instead, as noted by Geirhos et al. (2020), it is influenced by compounded learning phenomena related to which features are captured during training. Consequently, future research should account for these interactions when designing evaluation protocols. For example, one possibility is to test the model on *all* possible combinations of shift factors, thereby more comprehensively revealing how different combinations and sequences of shifts affect robustness.

**Model Selection**   While our measure has been primarily intended for evaluation, in this section, we investigate whether PA can also effectively guide *model selection*. To that end, we now examine a setting where the inductive bias is deliberately "poisoned" by shortcut opportunities (SO), *i.e.*, spurious correlations between the predicted factor $F_P$ and the learning factor $F_L$. We train a ResNet50 architecture from scratch using datasets $X'$ and $X''$, which differ only in the hue factor. These data sets are resampled for each shortcut-opportunity configuration, where spurious correlations between the predicted factor $F_P$ and the learning factor $F_L$ are systematically controlled. As illustrated in Figure 12 (and in more detail Appendix E), we systematically control these co-occurrences to create zero or partial Generalization Opportunities (GO). Under Zero-GO (ZGO), each instance of $F_P$ is exclusively paired with one instance of $F_L$, strongly encouraging models to overfit to the shortcut. Partial GOs (1/2/3-CGO) break some of these exclusive pairings, while Zero-SO (ZSO) allows all factor combinations. Generalization performance is then evaluated on progressively shifted test datasets, equivalent to those used in our previous experiment. Epoch-wise model selection is performed by either maximizing accuracy or maximizing PA on the validation sets, and the difference in

|  | Test 1 | | Test 2 | | Test 3 | | Test 4 | | Test 5 | |
|---|---|---|---|---|---|---|---|---|---|---|
|  | **Acc.** | **ΔAcc.** | **Acc.** | **ΔAcc.** | **Acc.** | **ΔAcc.** | **Acc.** | **ΔAcc.** | **Acc.** | **ΔAcc.** |
| **ERM** | | | | | | | | | | |
| ZGO | 53.2 | ±0.01 | 54.6 | ±0.01 | 55.7 | ±0.01 | 66.7 | ±0.01 | 66.6 | ±0.01 |
| 1-CGO | 62.9 | +9.5 | 64.7 | +10.2 | 60.8 | +0.3 | 62.9 | +2.2 | 64.2 | +0.5 |
| 2-CGO | 69.1 | +9.4 | 71.2 | +7.8 | 71.9 | +2.2 | 76.2 | -2.4 | 77.0 | -2.8 |
| 3-CGO | 73.1 | +16.6 | 85.6 | +3.6 | 70.1 | +9.7 | 71.4 | +6.4 | 72.1 | +6.7 |
| ZSO | 99.6 | ±0.01 | 92.8 | -0.1 | 89.9 | ±0.01 | 89.9 | +0.2 | 85.9 | ±0.01 |
| **IRM** | | | | | | | | | | |
| ZGO | 50.1 | +5.9 | 50.5 | +4.9 | 52.8 | +9.5 | 64.4 | +1.1 | 69.4 | +1.2 |
| 1-CGO | 63.0 | +7.0 | 65.9 | +7.6 | 59.4 | +2.2 | 59.0 | +1.8 | 59.0 | +1.8 |
| 2-CGO | 69.0 | +10.6 | 69.7 | +10.0 | 65.8 | +4.7 | 77.0 | +13.0 | 65.1 | +12.6 |
| 3-CGO | 79.5 | +11.6 | 83.0 | +9.8 | 73.6 | +10.9 | 79.5 | +11.0 | 72.2 | +11.3 |
| ZSO | 99.4 | +0.1 | 93.4 | +1.3 | 89.2 | +0.2 | 87.0 | +1.6 | 87.0 | +1.6 |

Table 1: Test performance under increasing levels of shift for models selected through different configurations of factor co-occurrence in the hue-based learning factor experiment. Specifically, the performance of models selected through validation accuracy (Acc.) and the difference between accuracy-based and PA-based selection (ΔAcc.) are reported. PA is able to select models that perform better than the accuracy-selected model in most cases.

generalization performance between these criteria is reported. For more details on the experiment setup, including the precise train/validation configurations, refer to Appendix E.

In contrast to the previous experiments, we exclude LISA here because its data-augmentation/interpolation strategies would undermine the carefully constructed SO/GO configurations as the algorithm augments the data based on the same image transformations present in this experiment. We therefore restrict our study to ERM and IRM, trained on source environments. Additionally, following the model-selection findings from earlier sections, we conduct in-distribution validation: the validation datasets match the same shift configuration as the training datasets, ensuring that model selection is not confounded by unseen shifts. The results in Table 1 show that *robustness-driven* model selection (based on PA) substantially improves the test performance, in particular, in settings with partial GOs (*i.e.*, 1-CGO or 3-CGO). In these cases, shortcut learning is partially mitigated, so identifying models that best maintain consistent predictions across slightly varied conditions (rather than just maximizing raw accuracy) proves crucial. Notably, IRM tends to benefit more from these robustness-based criteria, converging to models that can resist spurious correlations and better exploit the available GOs. Overall, this approach demonstrates how a focus on robust performance through PA, rather than pure in-distribution accuracy can significantly enhance generalization.

## 5 Discussion

The PA measure is an attempt at assessing robustness under an epistemologically grounded approach. Such an assessment is agnostic of the underlying data generation process and even the nature of the data itself, therefore, it is versatile and can be applied to explain diverse scenarios with a single theory, as illustrated in the experimental study. Additionally, PA is general in its scope. Our proposed version requires only the probabilistic outputs of the model, *e.g.*, the logits of a model, and no supervision. The model itself does not need to be differentiable, as we do not optimize its parameters.

By relying on the MEP, PA provides an estimate of a model's robustness which is as neutral as possible with respect to missing information. The measure relies on predictive confidence rather than predictions, thus providing a more fine-grained evaluation and a better discriminative assessment of model robustness. Indeed, the overlap between posteriors over the hypothesis space is indicative of the nature of the shift at an even greater detail than in standard analyses. Instead, test performance measures, such as $AFR_T$ rely solely

on hard-counting of "successes", therefore, they do not fulfill the desired properties of a robustness measure and are more prone to inconsistencies in the rankings.

In particular, in the adversarial learning setting, PA provides higher discriminability and consistency across the various covariate shifts, which is only partially matched by AFR in highly perturbed settings. Additionally, the analysis of the $\beta$ parameter provides further insights into the robustness of the models. In the domain generalization setting, PA further highlights that the robustness of the tested models decreases only with the first shifts, then it stabilizes. $\text{AFR}_T$, instead, keeps decreasing, showing once again that robustness and performance are two distinct concepts. In this setting, PA also proves to be very efficient in performing model selection.

Our proposed version of PA has, however, limitations. The theoretical framework has been developed by considering only the case of a finite hypothesis set. While this does not pose a severe drawback, and the experimental results attest to the validity of this approach, a characterization of PA for continuous hypothesis sets would be much more general in its scope. Additionally, we have assumed a uniform probability $p(c)$, to obtain a more tractable formula for the measure, ignoring the complexity of the hypothesis set. Adding this information would improve the robustness estimation process, allowing for more fine-grained discrimination between models. In the future, we plan to extend some of the presented theoretical results to more general kernels.

## 6 Conclusions

In this paper, we have set some desiderata for designing a measure measuring robustness against forms of covariate shift such as adversarial noise and domain generalization. Following theoretically grounded thinking, we proposed PA, a measure derived from the Posterior Agreement framework. We have conducted a comparison against commonly used accuracy measures, presenting analyses based on novel aspects of the shift. Our study shows that PA (i) provides a consistent and reliable robustness scoring, (ii) provides further details to the analysis that would be otherwise overlooked by accuracy measures and (iii) is also an efficient measure for model selection, while requiring no supervision. In conclusion, our work lays the basis for a more sound model robustness assessment, *in the PA sense.*

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

# A    Proofs of theorems

## A.1    Proof of Theorem 1

We first require the following lemma. For ease of reading, we slightly abuse the notation and consider $c$ to be a mapping from the object indices $i$ instead of the measurements $x_i$, that is $c : \{1, \ldots, N\} \to \{1, \ldots, K\}$.

**Lemma 1.** *Let $N, K \in \mathbb{N}$ and let $\{\mathcal{E}_{ij} \mid i \leq N, \ j \leq K\}$ be an indexed set of values. Then,*

$$\sum_{c \in \mathcal{C}} \prod_{i=1}^{N} \mathcal{E}_{i,c(i)} = \prod_{i=1}^{N} \sum_{j=1}^{K} \mathcal{E}_{ij} \tag{15}$$

*Proof.* By induction on $N$. For the $N = 1$ base case, observe that $\mathcal{C}$ has only $K$ elements, as there are only $K$ functions mapping $\{1\}$ to $\{1, \ldots, K\}$. Then

$$\sum_{c \in \mathcal{C}} \prod_{i \leq N} \mathcal{E}_{i,c(i)} = \sum_{c \in \mathcal{C}} \mathcal{E}_{1,c(1)} = \sum_{j \leq K} \mathcal{E}_{1,j} = \prod_{i \leq N} \sum_{j \leq K} \mathcal{E}_{i,j}. \tag{16}$$

Assume now that the result holds for some $N$. We demonstrate then that it also holds for $N + 1$. Observe that there is a bijection between $\mathcal{C}$ and $\{1, \ldots, K\}^N$. Therefore, we identify every function $c \in \mathcal{C}$ with the tuple $(c(1), \ldots, c(N))$. Conversely, we identify every tuple $(c_1, \ldots, c_N) \in \{1, \ldots, K\}^N$, with the function $c$ that maps $i$ to $c_i$.

$$
\begin{aligned}
\sum_{c \in \mathcal{C}} \prod_{i \leq N+1} \mathcal{E}_{i,c(i)} = \\
= \sum_{(c_1, \ldots, c_{N+1}) \in \{1, \ldots, K\}^{N+1}} \prod_{i \leq N+1} \mathcal{E}_{i,c_i} \\
= \sum_{\substack{(c_1, \ldots, c_N) \in \{1, \ldots, K\}^N \\ c_{N+1} \leq K}} \prod_{i \leq N+1} \mathcal{E}_{i,c_i} \\
= \sum_{(c_1, \ldots, c_N) \in \{1, \ldots, K\}^N} \sum_{c_{N+1} \leq K} \prod_{i \leq N+1} \mathcal{E}_{i,c_i} \\
= \sum_{(c_1, \ldots, c_N) \in \{1, \ldots, K\}^N} \sum_{c_{N+1} \leq K} \left( \mathcal{E}_{N+1,c(N+1)} \prod_{i \leq N} \mathcal{E}_{i,c_i} \right) \\
= \left( \sum_{c_{N+1} \leq K} \mathcal{E}_{N+1,c(N+1)} \right) \sum_{(c_1, \ldots, c_N) \in \{1, \ldots, K\}^N} \prod_{i \leq N} \mathcal{E}_{i,c_i} \\
= \left( \sum_{c_{N+1} \leq K} \mathcal{E}_{N+1,c(N+1)} \right) \prod_{i \leq N} \sum_{j \leq K} \mathcal{E}_{i,j} \\
= \left( \sum_{j \leq K} \mathcal{E}_{N+1,j} \right) \prod_{i \leq N} \sum_{j \leq K} \mathcal{E}_{i,j} \\
= \prod_{i \leq N+1} \sum_{j \leq K} \mathcal{E}_{i,j}.
\end{aligned}
\tag{17}
$$

$\square$

We are ready to prove the theorem

**Theorem 1**

$$p(c \mid X) = \prod_{i=1}^{N} p(c(x_i) \mid X), \tag{18}$$

where

$$p(k \mid X) = \frac{\exp(\beta F_k(x_i))}{\sum_{j=1}^{K} \exp(\beta F_j(x_i))} \tag{19}$$

is the probability that $x_i$ is assigned to class $k$.

*Proof.* The Gibbs distribution is

$$p(c \mid X) = \frac{\exp\left(\beta \sum_{i \leq N} F_{c(x_i)}(x_i)\right)}{\sum_{c \in \mathcal{C}} \exp\left(\beta \sum_{i \leq N} F_{c(x_i)}(x_i)\right)}. \tag{20}$$

The numerator can be rewritten as follows:

$$\exp\left(\beta \sum_{i \leq N} F_{c(x_i)}(x_i)\right) = \prod_{i \leq N} \exp\left(\beta F_{c(x_i)}(x_i)\right). \tag{21}$$

We now apply Lemma 1:

$$\sum_{c \in \mathcal{C}} \prod_{i \leq N} \exp\left(\beta F_{c(x_i)}(x_i)\right) = \prod_{i \leq N} \sum_{k \leq K} \exp\left(\beta F_k(x_i)\right) \tag{22}$$

Putting these results together yields that

$$p(c \mid \theta, X) = \frac{\prod_{i \leq N} \exp\left(\beta F_{c(x_i)}(x_i)\right)}{\prod_{i \leq N} \sum_{k \leq K} \exp\left(\beta F_k(x_i)\right)} \tag{23}$$

$$= \prod_{i \leq N} \frac{\exp\left(\beta F_{c(x_i)}(x_i)\right)}{\sum_{k \leq K} \exp\left(\beta F_k(x_i)\right)} \tag{24}$$

$$= \prod_{i \leq N} p(c(x_i) \mid X). \tag{25}$$

$\square$

## A.2  Proof of Theorem 2

**Theorem 2**  *With no prior information available, the empirical posterior agreement kernel $k(X', X'')$, can be rewritten as:*

$$k(X', X'') = \log\left(|C| \prod_{i=1}^{N} \sum_{j=1}^{K} p_i(j \mid X') p_i(j \mid X'')\right). \tag{26}$$

*Proof.*

$$k(X', X'') = \log\left(\sum_{c \in \mathcal{C}} \frac{p(c \mid X') p(c \mid X'')}{\pi(c)}\right)$$

$$= \log\left(|\mathcal{C}| \sum_{c \in \mathcal{C}} \prod_{i=1}^{N} p_i(c(x_i) \mid X') \prod_{i=1}^{N} p_i(c(x_i) \mid X'')\right)$$

$$= \log\left(|\mathcal{C}| \sum_{c \in \mathcal{C}} \prod_{i=1}^{N} p_i(c(x_i) \mid X') p_i(c(x_i) \mid X'')\right)$$

$$= \log\left(|\mathcal{C}| \prod_{i=1}^{N} \sum_{k=1}^{K} p_i(k \mid X') p_i(k \mid X'')\right). \tag{27}$$

The last step is obtained by applying Lemma 1. □

### A.3   Proof of Theorem 3

**Theorem 3**   *Under no prior information available, the following properties hold for the empirical posterior agreement kernel:*

1. *Boundedness: $0 \le PA(X', X'') \le \log K$.*

2. *Symmetry: $PA(X', X'') = PA(X'', X')$.*

3. *Concavity: $PA(X', X'')$ is a concave function in $\beta < +\infty$.*

*Proof.*

**Property 1 (Boundedness)**   When all predictions match, $\beta \longrightarrow +\infty$ and posteriors converge to Kronecker deltas centred at their respective MAPs, $\hat{y}_i'$ and $\hat{y}_i''$, coinciding for each $i = 1, \dots, N$:

$$\text{PA}(X', X''; \beta^*) = \frac{1}{N} \log \left( |C| \prod_{i=1}^{N} \sum_{j=1}^{K} \delta_{j\hat{y}_i'} \delta_{j\hat{y}_i''} \right) = \frac{1}{N} \log \left( |C| \prod_{i=1}^{N} 1 \right) = \frac{1}{N} \left( \log(|C|) + \sum_{i=1}^{N} \log(1) \right)$$

$$= \frac{\log(|C|)}{N} = \frac{\log(K^N)}{N} = \log(K).$$

When none of the predictions matches, $\beta \longrightarrow 0$ and posteriors converge to a uniform distribution:

$$\text{PA}(X', X''; \beta^*) = \frac{1}{N} \log \left( |C| \prod_{i=1}^{N} \sum_{j=1}^{K} \frac{1}{K} \cdot \frac{1}{K} \right) = \frac{1}{N} \log \left( |C| \prod_{i=1}^{N} \frac{1}{K} \right) = \frac{1}{N} \left( \log(|C|) + \sum_{i=1}^{N} \log(K^{-1}) \right)$$

$$= \frac{\log(|C|)}{N} - \log(K) = \log(K) - \log(K) = 0.$$

**Property 2 (Symmetry)**   Trivial, as it follows from the commutativity of the product.

**Property 3 (Concavity)**   First, note that

$$k\left(X', X''\right) = \log \left( |C| \prod_{i=1}^{N} \sum_{j=1}^{K} p_i\left(j \mid X'\right) p_i\left(j \mid X''\right) \right) \propto \sum_{i=1}^{N} \log \left( \sum_{j=1}^{K} p_i\left(j \mid X'\right) p_i\left(j \mid X''\right) \right), \quad (28)$$

where the posteriors $p_i(j \mid X)$, $X \in \{X', X''\}$ are the Gibbs distributions over classes for each observation[6]:

$$p_i(j \mid X) = \frac{\exp(\beta \mathcal{S}(j, x_i))}{\sum_{k=1}^{K} \exp(\beta \mathcal{S}(k, x_k))}. \quad (29)$$

Since the sum of concave functions is concave, we will focus on proving only the concavity of the log term. In particular, we will show that

$$\lambda(\beta) = -\log \left( \sum_{j=1}^{K} p(j \mid X') p(j \mid X'') \right) \quad (30)$$

is convex.

---

[6]Note that we do not specify the form of the cost function, therefore the theorem can be applied with any cost function $R(j, x_i)$.

Let us define $\mathcal{S}(j, x') = \mathcal{S}'_j$ and $\exp(\beta \mathcal{S}(j, x')) = \exp(\beta \mathcal{S}'_j) = e'_j$, with $x' \in X'$. Similar notation is used for $x'' \in X''$. $f(\beta)$ can be therefore rewritten as

$$\lambda(\beta) = -\log\left(\frac{\sum_{j=1}^K e'_j e''_j}{\sum_{k=1}^K e'_k \sum_{p=1}^K e''_p}\right) = -\log\left(\sum_{j=1}^K e'_j e''_j\right) + \log\left(\sum_{k=1}^K e'_k \sum_{p=1}^K e''_p\right) = -\lambda_1(\beta) + \lambda_2(\beta). \quad (31)$$

First, let us focus on the first term. In particular,

$$\frac{d}{d\beta}\lambda_1(\beta) = \frac{\sum_{j=1}^K (\mathcal{S}'_j + \mathcal{S}''_j) e'_j e''_j}{\sum_{k=1}^K e'_k e''_k}. \quad (32)$$

The second derivative is

$$\frac{d^2}{d\beta^2}\lambda_1(\beta) = \frac{\sum_{j=1}^K (\mathcal{S}'_j + \mathcal{S}''_j)^2 e'_j e''_j}{\sum_{k=1}^K e'_j e''_j} - \left(\frac{\sum_{j=1}^K (\mathcal{S}'_j + \mathcal{S}''_j) e'_j e''_j}{\sum_{k=1}^K e'_j e''_j}\right)^2. \quad (33)$$

Therefore,

$$\frac{d^2}{d\beta^2}\lambda_1(\beta) > 0 \iff \left(\sum_{k=1}^K e'_j e''_j\right)\left(\sum_{j=1}^K (\mathcal{S}'_j + \mathcal{S}''_j)^2 e'_j e''_j\right) - \left(\sum_{j=1}^K (\mathcal{S}'_j + \mathcal{S}''_j) e'_j e''_j\right)^2 > 0 \quad (34)$$

Using the distributive property of the product over the sum, the expression becomes

$$\sum_{k=1}^K \sum_{j=1}^K (\mathcal{S}'_j + \mathcal{S}''_j)^2 e'_j e''_j e'_k e''_k - \sum_{k=1}^K \sum_{j=1}^K (\mathcal{S}'_j + \mathcal{S}''_j)(\mathcal{S}'_k + \mathcal{S}''_k) e'_j e''_j e'_k e''_k > 0 \quad (35)$$

$$\iff \sum_{k=1}^K \sum_{j=1}^K [(\mathcal{S}'_j + \mathcal{S}''_j) - (\mathcal{S}'_k + \mathcal{S}''_k)](\mathcal{S}'_j + \mathcal{S}''_j) e'_j e''_j e'_k e''_k > 0 \quad (36)$$

Let $\Delta_{(jj),(kk)} = (\mathcal{S}'_j + \mathcal{S}''_j) - (\mathcal{S}'_k + \mathcal{S}''_k)$ define the difference in the cost attributed to reference class $j$ and the cost attributed to class $k$, accumulated over $X', X''$, and $E_{jk} = e'_j e''_j e'_k e''_k$.

Overall, the sum can be expressed as:

$$\sum_{k=1}^K \sum_{j=1}^K [(\mathcal{S}'_j + \mathcal{S}''_j) - (\mathcal{S}'_k + \mathcal{S}''_k)](\mathcal{S}'_j + \mathcal{S}''_j) e'_j e''_j e'_k e''_k = \sum_{k=1}^K \sum_{j=1}^K (\mathcal{S}'_j + \mathcal{S}''_j) E_{jk} \Delta_{(jj),(kk)} = \sum_{k=1}^K \sum_{j=1}^K G_{(jj),(kk)} \quad (37)$$

Note that $\Delta_{(jj),(jj)} = 0 \implies G_{(jj),(jj)} = 0$. Moreover, $\Delta_{(jj),(kk)} = -\Delta_{(kk),(jj)}$ and $E_{jk} = E_{kj}$. Then, the previous term can be rewritten as

$$\sum_{j=1}^K \sum_{k<j}^K G_{(jj),(kk)} + G_{(kk),(jj)} = \sum_{j=1}^K \sum_{k<j}^K (\mathcal{S}'_j + \mathcal{S}''_j) E_{jk} \Delta_{(jj),(kk)} + (\mathcal{S}'_k + \mathcal{S}''_k) E_{kj} \Delta_{(kk),(jj)} \quad (38)$$

$$= \sum_{j=1}^K \sum_{k<j}^K E_{jk} \Delta_{(jj),(kk)}[(\mathcal{S}'_j + \mathcal{S}''_j) - (\mathcal{S}'_k + \mathcal{S}''_k)] = \sum_{j=1}^K \sum_{k<j}^K E_{jk} \Delta^2_{(jj),(kk)} \quad (39)$$

The last term is strictly positive for $E_{jk} > 0 \implies e'_j, e''_j > 0$, for $j = 1, \ldots, K$, which is always possible for $\beta > 0$. Therefore, the first term is convex.

We proceed equivalently with the second term:

$$\lambda_2(\beta) = \log\left(\sum_{j=1}^K e'_j \sum_{k=1}^K e''_k\right) = \log\left(\sum_{k=1}^K \sum_{j=1}^K e'_j e''_k\right) \quad (40)$$

$$\frac{d}{d\beta}\lambda_2(\beta) = \frac{\sum_{k=1}^{K}\sum_{j=1}^{K}(\mathcal{S}_j' + \mathcal{S}_k'')e_j'e_k''}{\sum_{k=1}^{K}\sum_{j=1}^{K}e_j'e_k''} \tag{41}$$

$$\frac{d^2}{d\beta^2}\lambda_2(\beta) = \frac{\sum_{k=1}^{K}\sum_{j=1}^{K}(\mathcal{S}_j' + \mathcal{S}_k'')^2 e_j'e_k''}{\sum_{k=1}^{K}\sum_{j=1}^{K}e_j'e_k''} - \left(\frac{\sum_{k=1}^{K}\sum_{j=1}^{K}(\mathcal{S}_j' + \mathcal{S}_k'')e_j'e_k''}{\sum_{k=1}^{K}\sum_{j=1}^{K}e_j'e_k''}\right)^2 > 0 \tag{42}$$

$$\iff \left(\sum_{k=1}^{K}\sum_{j=1}^{K}e_j'e_k''\right)\left(\sum_{k=1}^{K}\sum_{j=1}^{K}(\mathcal{S}_j' + \mathcal{S}_k'')^2 e_j'e_k''\right) - \left(\sum_{k=1}^{K}\sum_{j=1}^{K}(\mathcal{S}_j' + \mathcal{S}_k'')e_j'e_k''\right)^2 > 0 \tag{43}$$

$$\iff \sum_{k=1}^{K}\sum_{q=1}^{K}\sum_{j=1}^{K}\sum_{i=1}^{K}(\mathcal{S}_j' + \mathcal{S}_k'')^2 e_j'e_k''e_i' - (\mathcal{S}_j' + \mathcal{S}_k'')e_j'e_k''(\mathcal{S}_i' + \mathcal{S}_q'')e_i'e_i'' > 0 \tag{44}$$

$$\iff \sum_{k=1}^{K}\sum_{q=1}^{K}\sum_{j=1}^{K}\sum_{i=1}^{K}(\mathcal{S}_j' + \mathcal{S}_k'')e_j'e_k''e_i'e_i''[(\mathcal{S}_j' + \mathcal{S}_k'') - (\mathcal{S}_i' + \mathcal{S}_q'')] > 0 \tag{45}$$

Similarly, as we did before, we define $e_j'e_k''e_i'e_i'' = E_{(jk),(iq)} = E_{(ik),(jq)} = E_{(jq),(ik)} = E_{(iq),(jk)}$ and $\Delta_{(jk),(iq)} = (\mathcal{S}_j' - \mathcal{S}_i') + (\mathcal{S}_k'' - \mathcal{S}_q'') = -\Delta_{(iq),(jk)}$. Then,

$$\frac{d^2}{d^2\beta}\lambda_2(\beta) = \sum_{k=1}^{K}\sum_{q=1}^{K}\sum_{j=1}^{K}\sum_{i=1}^{K}G_{(jk),(iq)} = \sum_{k=1}^{K}\sum_{q=1}^{K}\sum_{j=1}^{K}\sum_{i=1}^{K}(\mathcal{S}_j' + \mathcal{S}_k'')E_{(jk),(iq)}\Delta_{(jk),(iq)} \tag{46}$$

Therefore,

$$\sum_{k=1}^{K}\sum_{q<k}^{K}\sum_{j=1}^{K}\sum_{i<j}^{K}G_{(jk),(iq)} + G_{(iq),(jk)} \tag{47}$$

$$= \sum_{k=1}^{K}\sum_{q<k}^{K}\sum_{j=1}^{K}\sum_{i<j}^{K}(\mathcal{S}_j' + \mathcal{S}_k'')E_{(jk),(iq)}\Delta_{(jk),(iq)} + (\mathcal{S}_i' + \mathcal{S}_q'')E_{(iq),(jk)}\Delta_{(iq),(jk)} \tag{48}$$

$$= \sum_{k=1}^{K}\sum_{q<k}^{K}\sum_{j=1}^{K}\sum_{i<j}^{K}E_{(jk),(iq)}\Delta_{(jk),(iq)}[(\mathcal{S}_j' + \mathcal{S}_k'') - (\mathcal{S}_i' + \mathcal{S}_q'')] \tag{49}$$

$$= \sum_{k=1}^{K}\sum_{q<k}^{K}\sum_{j=1}^{K}\sum_{i<j}^{K}E_{(jk),(iq)}\Delta^2_{(jk),(iq)} \tag{50}$$

The last term is again positive for $e_j', e_j'' > 0$, $j = 1, \dots, K$.

Last, we prove the convexity of $\lambda(\beta)$:

$$\frac{d^2}{d\beta^2}\lambda(\beta) = \left(\sum_{k=1}^{K}\sum_{q<k}^{K}\sum_{j=1}^{K}\sum_{i<j}^{K}E_{(jk),(iq)}\Delta^2_{(jk),(iq)} - \sum_{j'=1}^{K}\sum_{i'<j'}^{K}E_{(j'j'),(k'k')}\Delta^2_{(j'j'),(k'k')}\right) \tag{51}$$

By reindexing $k' = j$ and $j' = i$ it is clear that the second sum is contained in the first one, thus the negative terms nullify, and the derivative is positive. Proving that $\lambda(\beta)$ is absolutely convex in $\mathbb{R}^+$. $\qquad\square$

Note that concavity is assured for positive $\beta$ and on the limit $\beta \to \infty$ the curvature is not defined, so it is advisable to start a numerical optimization procedure at a value $\beta_0 = 0^+$, since

$$\lim_{\beta \to 0^+}\frac{d^2}{d\beta^2}F(\beta) > 0. \tag{52}$$

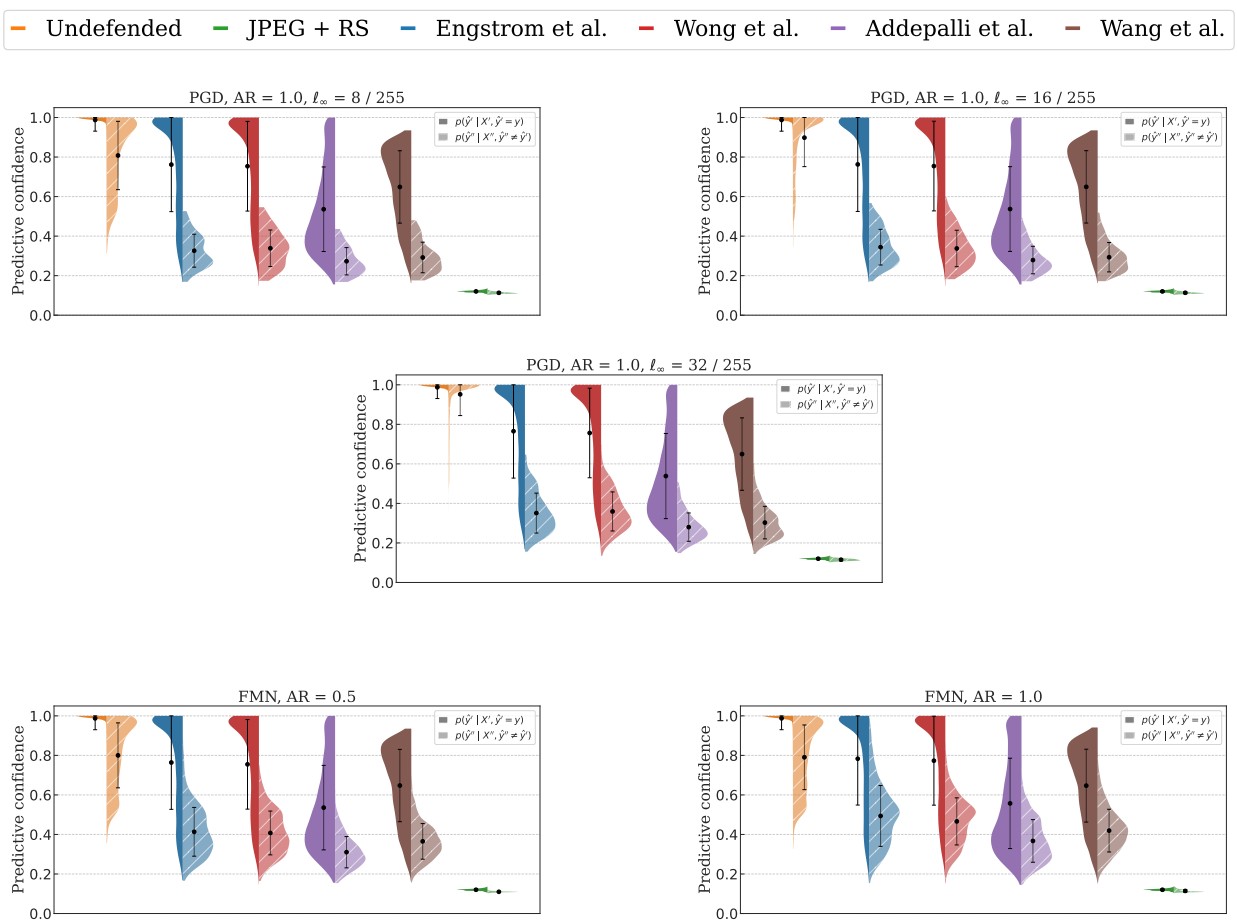

Figure 9: Adversarial setting: predictive confidence distribution for all tested models.

## B  Extended Results

In Figure 9, we report the predictive confidence distribution for all the tested models in the adversarial setting. Again, it can be seen that the robust models lower their average confidence on $X''$, showing that they are effectively detecting an ongoing attack. Addepalli et al. (2022) presents a long-tailed distribution before the attack, favouring more conservative, less confident predictions, which penalize the detection of an attack.

We include a plot of FMN for $AR = 1$. This case contains attacked images with a very high norm ($\ell_\infty > 32$). The models were not trained to resist such powerful attacks, therefore the average conditional confidence is more variable and all standard errors overlap.

## C  PA is not accuracy

It may be tempting to use Posterior Agreement to measure model performance instead of accuracy measures. In Figure 10, we show an example of why this is a wrong use of PA. In particular, we display the performance of PA and of a classification model (DistilBERT-base) under three different shift strategies: (i) Levenshtein: addition, removal or substitution of characters in the sentence, (ii) Amplification: addition of adjectives reinforcing the sentence's sentiment (iii) Contradiction: addition of adjective weakening the sentence's sentiment. While in cases (i) and (iii) both measures behave similarly, in case (ii) the addition of reinforcing adjectives has the effect of increasing the confidence in the model predictions, improving the

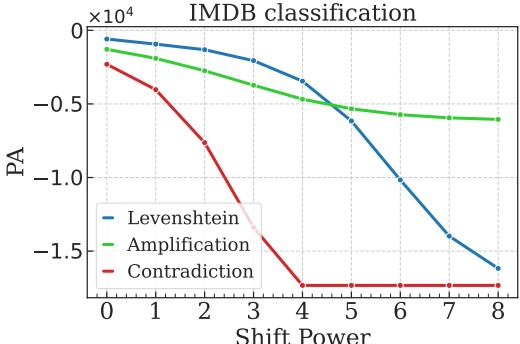 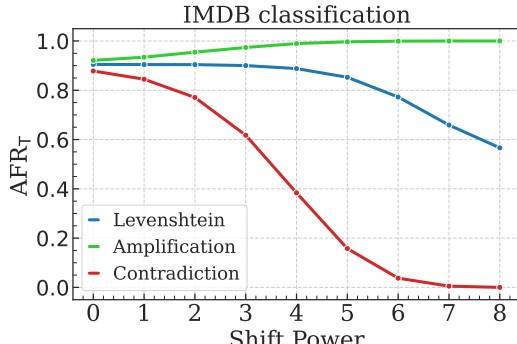

Figure 10: PA and accuracy for the IMDB sentiment classification task (Maas et al., 2011) under simple attacks. Observations are perturbed by manipulating some characters (Levenshtein) and by replacing some words with positive or negative adjectives that either encourage (Amplification) or discourage (Contradiction) the true sentiment of the review. The shift power is defined as $W$, being $2^W$ the number of replacements.

overall performance. The sensibility of the method to this shift is correctly detected by PA, that *penalizes* the model decreasing its robustness score. This example again remarks that PA and accuracy are different measures meant to investigate distinct aspects of a model.

## D  Robustness and feature alignment

To better understand the impact of controlled domain shifts on robustness assessment through PA, we used ERM and IRM algorithms to train a ResNet18 model for 50 epochs on $\mathcal{D}_{\mathrm{train}}$, using Adam with a learning rate of $10^{-2}$. From each validation sample $x_0^{\mathrm{val}}, x_1^{\mathrm{val}}$, we selected 128 observations to form a reduced validation set $\mathcal{D}_{\mathrm{sub}} = \{x_0^{\mathrm{sub}}, x_1^{\mathrm{sub}}\} \subset \mathcal{D}_{\mathrm{val}}$. Both $x_0^{\mathrm{sub}}$ and $x_1^{\mathrm{sub}}$ share the same random seed $\tau^{\mathrm{val}}$, thus containing the same MNIST samples in the same order, and they differ only by a hue-based distribution shift (blue *vs* red; *cf.* Table 2).

At the end of each epoch, we compute the principal component (via PCA) of the feature representations of $x_0^{\mathrm{sub}}$ and $x_1^{\mathrm{sub}}$ separately. Each MNIST observation in $\mathcal{D}_{\mathrm{sub}}$ thus has two projected values, one for each sample. By examining this principal component (the direction of greatest variance), we can qualitatively evaluate the inductive bias encoded after each training epoch.

Let $\Phi^c$ be the feature extractor of a classifier $c$ after a given epoch. Denote by $\mathbf{v}_0$ and $\mathbf{v}_1$ the principal directions of the feature space for $x_0^{\mathrm{sub}}$ and $x_1^{\mathrm{sub}}$, respectively. We compute the projections of each observation $x_{0,n}^{\mathrm{sub}}$ and $x_{1,n}^{\mathrm{sub}}$ onto these directions:

$$z_{0,n} \;=\; \langle \Phi^c(x_{0,n}^{\mathrm{sub}}), \mathbf{v}_0 \rangle, \quad z_{1,n} \;=\; \langle \Phi^c(x_{1,n}^{\mathrm{sub}}), \mathbf{v}_1 \rangle, \quad n = 1, \ldots, N_{\mathrm{sub}}.$$

The distribution of $z_{0,n}$ and $z_{1,n}$ across different classes indicates how well the inductive bias aligns with task-relevant features: ideally, class membership should drive the principal variance in the feature space. The class-conditional variance of these projections thus gauges the discriminative strength of the model's latent features and, in turn, its robustness to sampling variability.

Figure 11 illustrates the principal component projections of the feature space representations of samples $x_0^{sub}, x_1^{sub}$ for ERM and IRM algorithms at three different training stages. These results show that ERM is unable to encode a representation that is both discriminative and invariant to domain shifts, as it either displays a high cross-domain error or a high class-conditional variance. This indicates that its inductive bias is exclusively driven by domain-specific features or by class-specific features, and possibly the high learning rate avoids the model to converge to a more robust solution. In contrast, IRM is able to encode a representation that reduces both qualitative measures at the same time, which indicates that the inductive

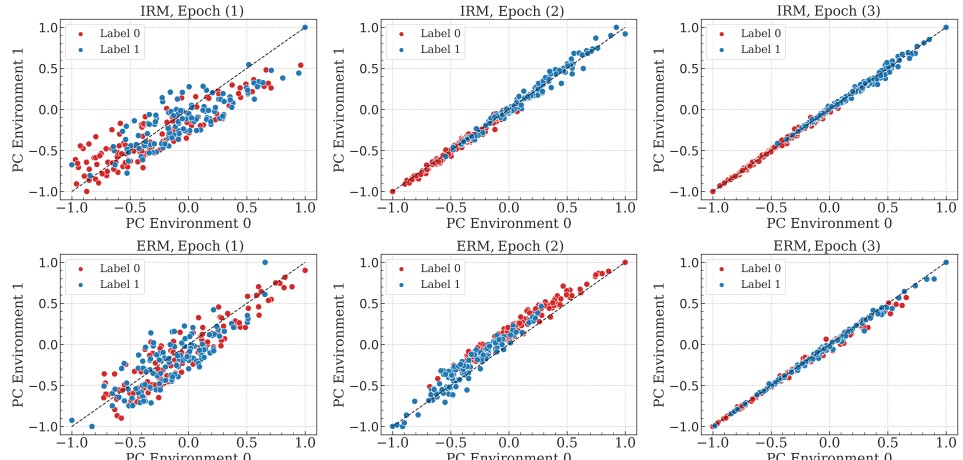

Figure 11: Normalized principal component for each environment at three different training stages. ERM (top) and IRM (bottom) algorithms are considered, and projections are colored by class membership. ERM projections display either a high cross-domain error or a high class-conditional variance. In contrast, IRM is able to encode a representation that reduces both measures at the same time, thus indicating a more robust inductive bias.

bias is able to capture the most predictive features for the task at hand without being significantly influenced by the shift in the hue factor.

These measures have been shown to qualitatively assess the suitability of the inductive bias for the aforementioned sources of randomness separately.

## E   Details on the DiagViB-6 Dataset

The DiagViB-6 dataset comprises both source domains ($\mathcal{S} = \{X_0, X_1\}$) and target domains ($\mathcal{T} = \{X_2, X_3, X_4, X_5\}$). Here, $X_j$ represents the random variable corresponding to domain $j$, where $j$ indicates the number of shifted factors relative to $X_0$. The classification task focuses on predicting the *shape factor* (the digit) using handwritten 4s and 9s from MNIST. We systematically control the *covariate shift* by varying visual attributes (*e.g.*, color/hue) while keeping class identities intact.

**Data Splits and Notation**   For each domain $X_j$, we generate four disjoint subsets of MNIST by applying different random instantiations $\tau_0^{\text{train}}$, $\tau_1^{\text{train}}$, $\tau^{\text{val}}$, and $\tau^{\text{test}}$. We thus define:

- $\mathcal{D}_{\text{train}} = \{x_0^{\text{train}}, x_1^{\text{train}}\}$, where $x_j := x_j \circ \tau_j^{\text{train}}$, for $j = 0, 1$.

- $\mathcal{D}_{\text{val}} = \{x_0^{\text{val}}, x_1^{\text{val}}\}$, where $x_j := x_j \circ \tau^{\text{val}}$, for $j = 0, 1$.

- $\mathcal{D}_{\text{test}}^{(j)} = \{x_j^{\text{test}}\}$, where $x_j^{\text{test}} := x_j^{\text{test}} \circ \tau^{\text{test}}$, for $j = 0, \ldots, 5$.

In this way, the *training* data is subject to both sampling randomness ($\tau_0^{\text{train}} \neq \tau_1^{\text{train}}$) and domain shift ($X_0 \not\sim X_1$), while the *validation* and *test* splits each use a single instantiation, ensuring that *distribution shift* (*i.e.*, changes in hue or other visual attributes) is the only significant source of randomness across $X'$ and $X''$. Overall, we create two sets of 40 000 images for training, two sets of 20 000 images for validation, and six sets of 10 000 images for testing.

**Controlled Covariate Shift**   By design, each shift factor introduced in domain $X_j$ modifies specific image features (*e.g.*, color channels, texture patterns) while preserving the underlying digit shape. Because each domain relies on the same MNIST base classes (4s and 9s), domain shift can be precisely *engineered* to test a

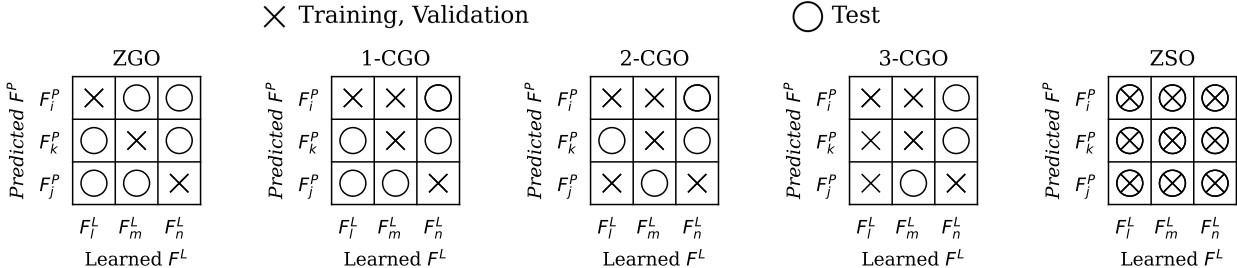

Figure 12: Representation of the co-occurrence pattern in between learning factors $F^L$ and predicted factors $F^P$ for the ZGO, CGO and ZSO settings that will be considered in this experiment.

| # Shift Factors | 0 | 1 | 2 | 3 | 4 | 5 |
|---|---|---|---|---|---|---|
| Hue | red | **blue** | blue | blue | blue | blue |
| Lightness | dark | dark | **bright** | bright | bright | bright |
| Position | CC | CC | CC | **LC** | LC | LC |
| Scale | normal | normal | normal | normal | **large** | large |
| Texture | blank | blank | blank | blank | blank | **tiles** |
| *Shape* | *4,9* | *4,9* | *4,9* | *4,9* | *4,9* | *4,9* |

Table 2: Specific image factors associated with each environment considered in the model discriminability experiments. CC and LC account for *centered center* and *centered low*, respectively.

model's robustness. Hence, the DiagViB-6 setup allows for a clear separation between *sampling randomness* and *shift-induced variability*:

- Sampling randomness: Different subsets of MNIST (different seeds) across training, validation, and test.

- Domain shift: Controlled variations (*e.g.*, hue) systematically applied to form $X_1, \ldots, X_5$ from $X_0$.

**Rationale for DiagViB-6**   This setup maximizes the potential for learning *invariant features* by exposing models to progressively more challenging shifts during training. It also provides strong *validation and testing* conditions to evaluate how well the learned representations generalize to new shifts. Because the same underlying digit classes are maintained across domains, the impact of domain shift on classification can be directly attributed to feature manipulation (*e.g.*, hue, stroke thickness), rather than confounded by changes in class identity. This approach follows the methodology outlined in Geirhos et al. (2020), where controlling image factors enables a clearer view of the *shortcut learning* phenomena.

**DiagViB-6 for Model Discriminability**   To adjust DiagViB-6 to specific parameters of our multiple environments, we have defined the values described in Table 2 as the shift factor settings for each environment. An example of training, validation, and test samples subject to different distribution shifts is depicted in Figure 13.

**DiagViB-6 for Model Selection**   To evaluate model selection under domain generalization, we examined the impact of *hue shift* as the primary source of inductive bias. Since hue represents a significant variation in image representation, this experiment provides insight into the consistency of model selection criteria. For this experiment, two different sampling instantiations for the validation datasets are considered, denoted as $\tau_0^{\text{val}} \neq \tau_1^{\text{val}}$. This means that $\mathcal{D}_{\text{val}}^{(0)}$ and $\mathcal{D}_{\text{val}}^{(1)}$ correspond to different instantiations of the hue shift factor. The training and validation dataset configurations are detailed in Table 2. Similarly to the model discriminability experiment, training datasets are always drawn from source domains and consist of samples $\mathcal{D}_{\text{train}} = \{x_0^{\text{train}}, x_1^{\text{train}}\}$. Validation datasets can be drawn from either source or target domains, leading to different generalization scenarios.

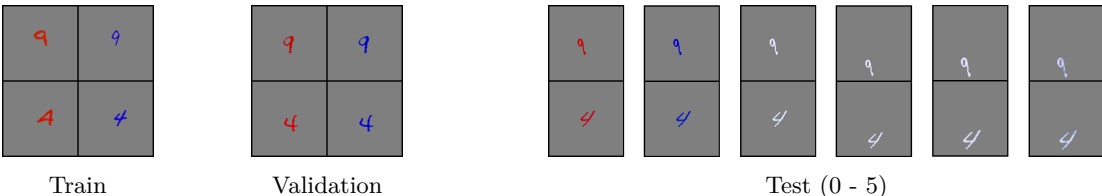

Figure 13: Samples from the training, validation and test datasets. Training samples belong to different MNIST subsets, whereas validation and test samples are the transformations of the same observation.

|  | Env. | Hue | Lightness | Position | Scale | Texture | *Shape* |
|---|---|---|---|---|---|---|---|
| Training | 0 | red | dark | CC | large | blank | *1,4,7,9* |
|  | 1 | **blue** | dark | CC | large | blank | *1,4,7,9* |
| Validation | 0 | red | dark | CC | large | blank | *1,4,7,9* |
| SD | 1 | red | dark | CC | large | blank | *1,4,7,9* |
| ID | 1 | **blue** | dark | CC | large | blank | *1,4,7,9* |
| 1F-MD | 1 | **magenta** | dark | CC | large | blank | *1,4,7,9* |
| 5F-MD | 1 | **green** | **bright** | **UL** | **small** | **tiles** | *1,4,7,9* |
| Validation OOD | 0 | **yellow** | dark | CC | large | blank | *1,4,7,9* |
|  | 1 | **magenta** | dark | CC | large | blank | *1,4,7,9* |

Table 3: Image factors associated with each of the environments considered in the model selection experiment. CC and UL account for 'centered center' and 'upper left', respectively.

First, when both $\mathcal{D}_{\text{val}}^{(0)}$ and $\mathcal{D}_{\text{val}}^{(1)}$ are drawn from *source domains*, two configurations are considered. If validation samples originate from the same distribution as training, they are in the *same distribution* (SD) setting. If they come from source domains but with different hue instantiations than the training data, they are in the *in-distribution* (ID) setting. Second, when $\mathcal{D}_{\text{val}}^{(0)}$ is drawn from source domains and $\mathcal{D}_{\text{val}}^{(1)}$ from target domains, the *mixed distribution* (MD) setting applies. Depending on the magnitude of the hue shift, we define two cases: a 1-*factor mixed distribution* (1F-MD), where only hue varies, and a 5-*factor mixed distribution* (5F-MD), where hue is combined with additional variations. Finally, when both $\mathcal{D}_{\text{val}}^{(0)}$ and $\mathcal{D}_{\text{val}}^{(1)}$ are drawn from *target domains*, the setup corresponds to the *out-of-distribution* (OOD) setting, where validation samples differ entirely from training.

This setup ensures that models are systematically tested under increasing levels of domain shift, from mild (ID, 1F-MD) to severe (5F-MD, OOD). By doing so, we assess the robustness of model selection criteria when generalizing to unseen hue shifts. For more details on the final dataset configuration, please refer to Table 3.

## F  Autoattack results

In order to test our measure on more realistic benchmarks, we include an analysis of PA vs $\text{AFR}_T$ using the well-known AutoAttack library (Croce & Hein, 2020b).

### F.1  CIFAR10

We test the previous models on the CIFAR-10 dataset under three attacks: APGD-CE & DLR (Croce & Hein, 2020b) and FAB Croce & Hein (2020a), all with $\ell_\infty = \ell/255$, $\ell \in [2, 4, 8, 16, 32]$. These attacks are significantly more effective than PGD and FMN, therefore, we are able to score robustness in the complete $0 - 100\%$ $\text{AFR}_T$ range.

In Figure 14-16, we can see the results for the attacks. The trends of the models are similar to the ones presented in the main paper, with $\text{AFR}_T$ failing to single out the undefended model. On the contrary, in the

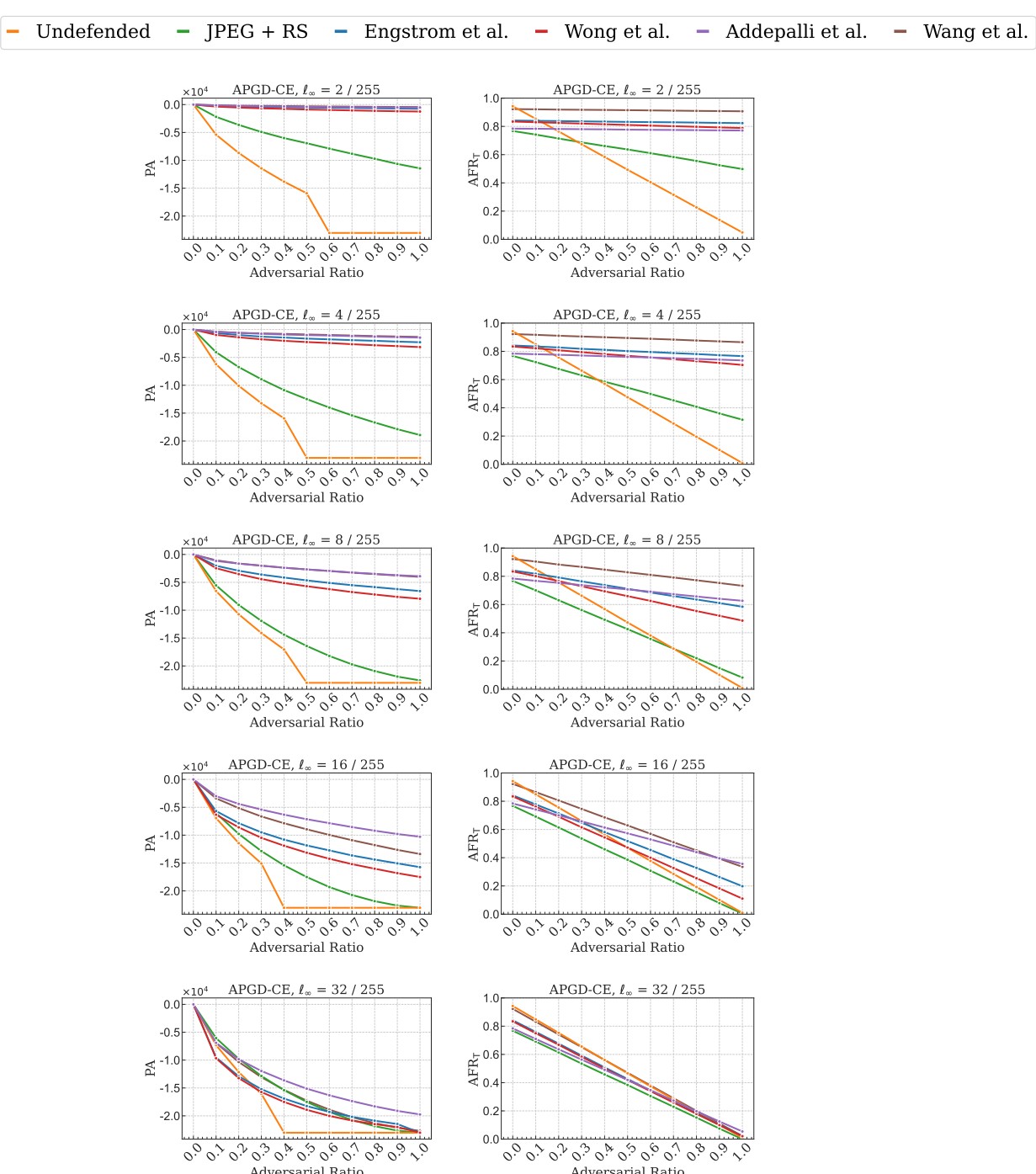

Figure 14: PA (left) and $\text{AFR}_T$ (right) scores against increasing AR and $\ell_\infty$, for the APGD-CE attack of the Autoattack library.

APGD family, PA scores its minimum already with $AR \ll 1$. Another notable phenomenon is represented by Addepalli et al. (2022) model, which is scored as the best or second-best model according to PA. In $\text{AFR}_T$, this happens only in the presence of large $AR$ and $\ell_\infty$ values, due to the faster worsening of the other models. This highlights once again the risk of inconsistency in the robustness evaluation when using $\text{AFR}_T$,

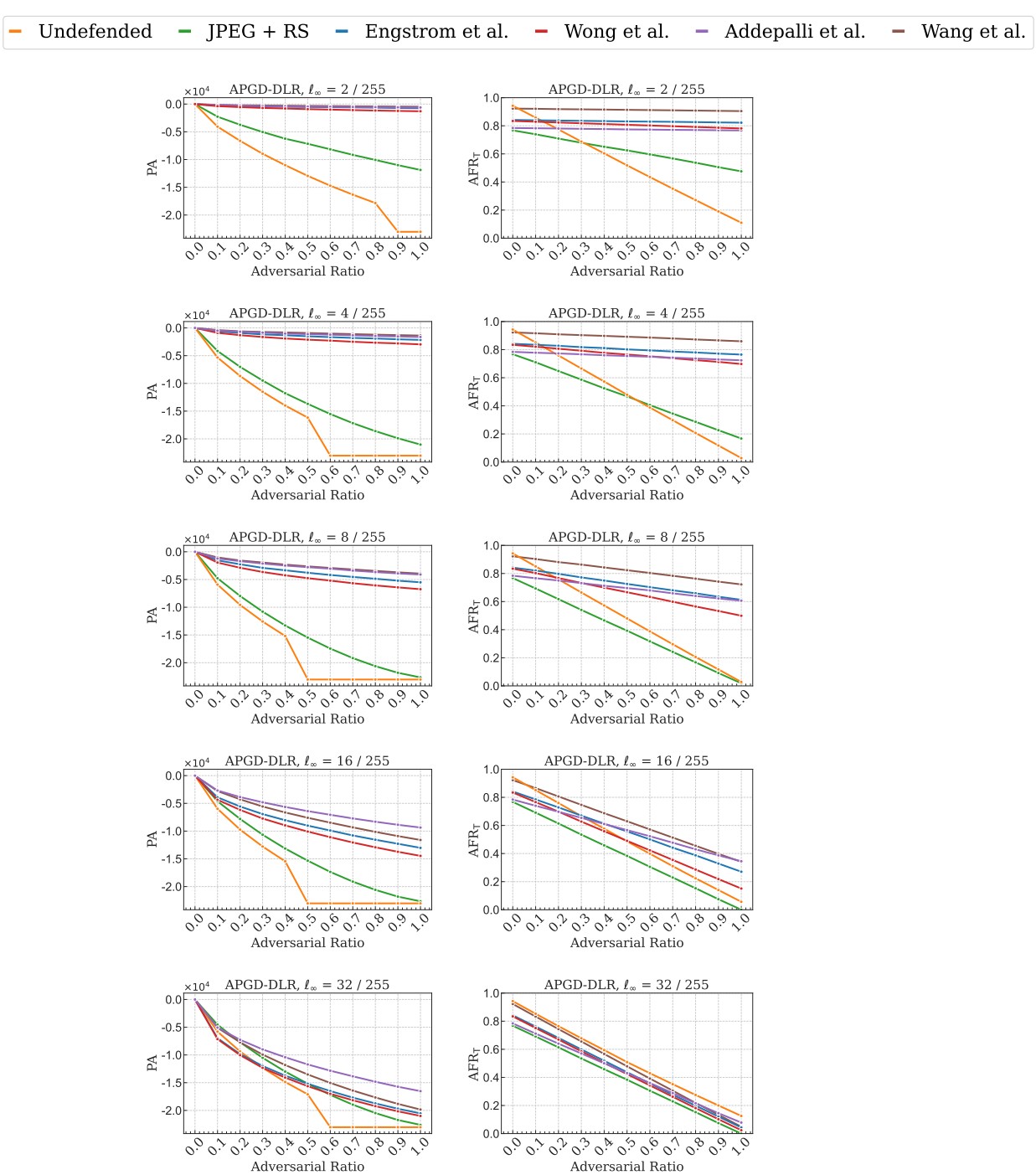

Figure 15: PA (left) and $\text{AFR}_T$ (right) scores against increasing AR and $\ell_\infty$, for the APGD-DLR attack of the Autoattack library.

while PA maintains an overall stable ranking across the $\ell_\infty$ values. As witnessed with PGD and FMN, PA correctly estimates the models' performance and ranks them as $\text{AFR}_T$ ($AR = 0$) does (this is not easily noticeable in the plots due to the scale used).

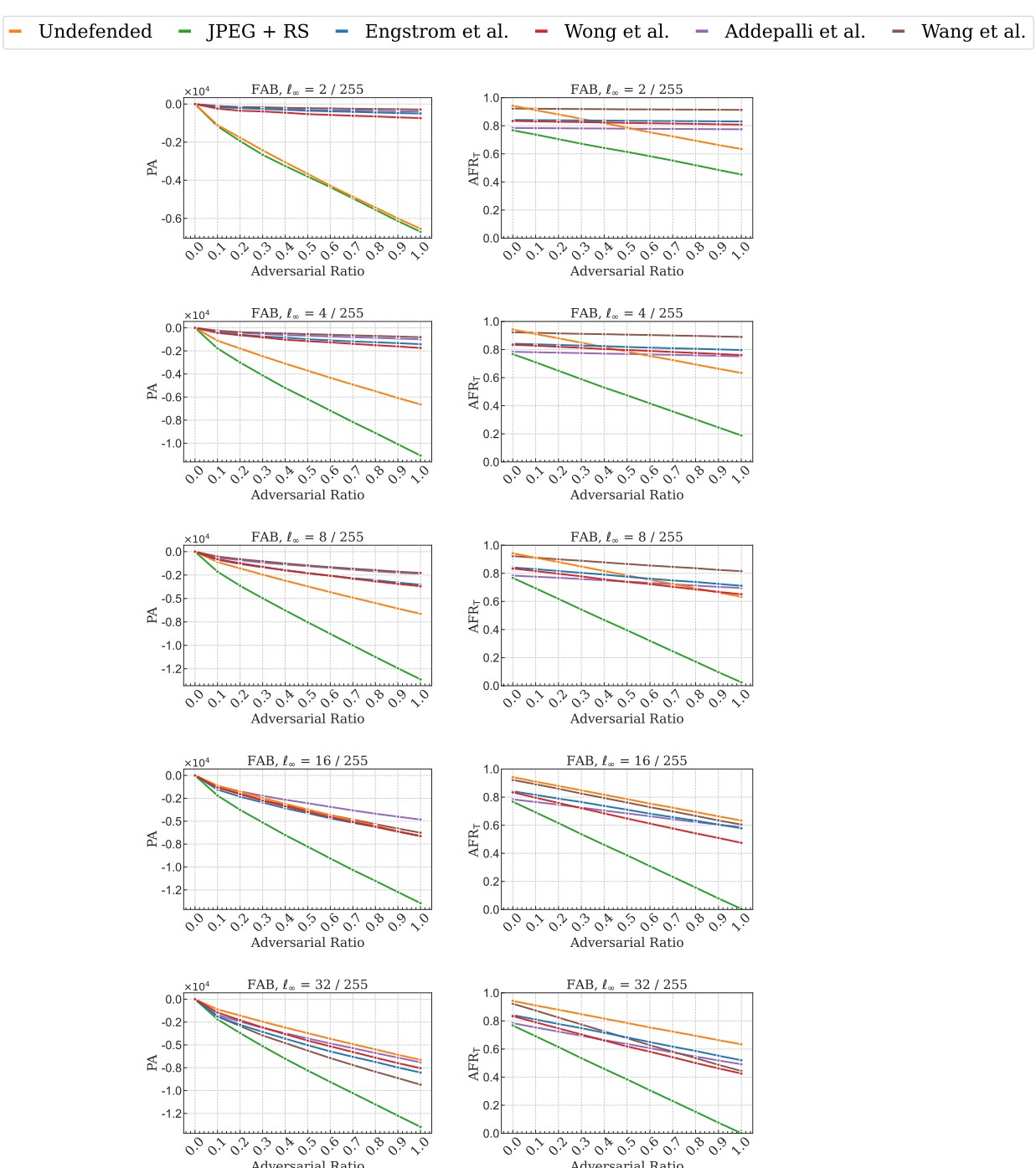

Figure 16: PA (left) and AFR$_T$ (right) scores against increasing AR and $\ell_\infty$, for the FAB attack of the Autoattack library.

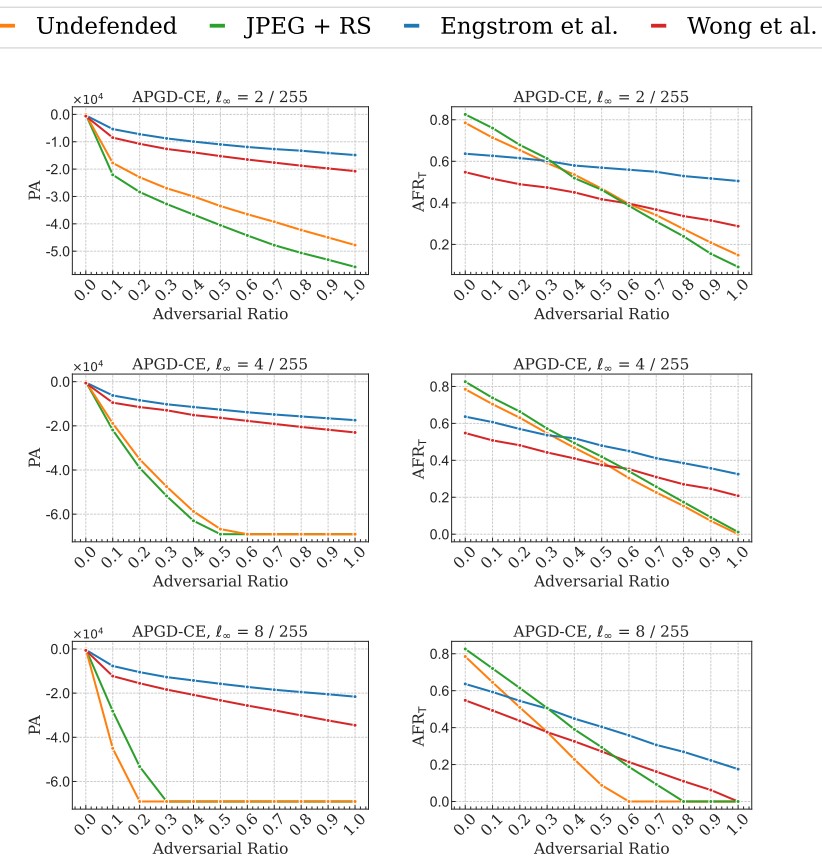

Figure 17: PA (left) and AFR$_T$ (right) scores against increasing AR and $\ell_\infty$, for the APGD-CE attack of the Autoattack library.

## F.2 ImageNet

We select four of the fastest models and test them on the ImageNet dataset. Following (Croce & Hein, 2020b), we use the APGD-CE & APGD-DLR$^\top$, an approximated, faster version of APGD-DLR for large datasets. We test with $\ell_\infty = \ell/255$, $\ell \in [2, 4, 8]$.

In Figure 17-18, we report the results. We obtain similar results to those in the previous case, with PA discriminating the weak models from the robust ones. In this case, JPEG+RS defense is only resisting the attack at higher power, and behaves as undefended for lower ones. This may be due to the different confidence in the outputs, on which the efficiency of the method depends. Otherwise, PA provides a robustness assessment that aligns with AFR$_T$ at high $AR$ values, showcasing its increased discriminability.

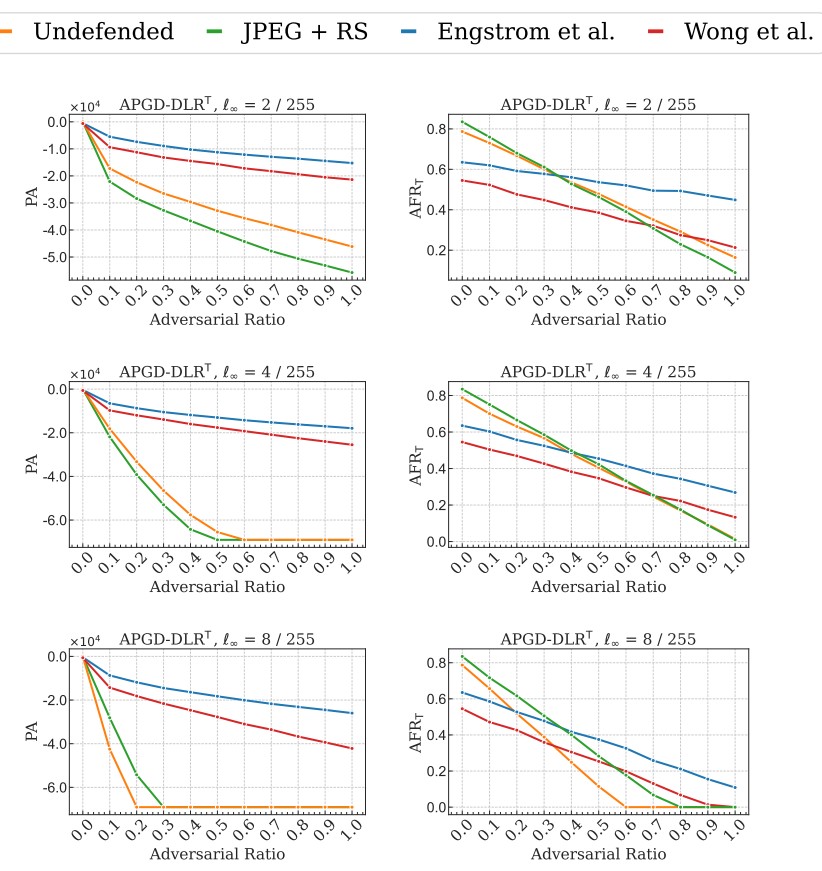

Figure 18: PA (left) and $\text{AFR}_T$ (right) scores against increasing AR and $\ell_\infty$, for the targeted APGD-DLR$^\top$ attack of the Autoattack library.

