# OpenReview forum: "Rethinking Robustness in Machine Learning: A Posterior Agreement Approach"
_TMLR — Accepted by TMLR_

### Review · Reviewer_iaqh · 2025-05-18

**Summary Of Contributions:**

This paper studies the problem of evaluating the robustness of machine learning models under covariate shifts, including adversarial shift and out-of-distribution shift. The authors argue that traditional metrics like accuracy-based scores are insufficient for assessing robustness, because the authors claim that the metric should differentiate models only by their generalization capabilities against covariate shift and should be independent of the task performance of the model.

To overcome this, they introduce a novel metric called Posterior Agreement (PA), grounded in information theory and the Maximum Entropy Principle. The PA metric quantifies the agreement between posterior distributions of model predictions under different data distributions, providing a more nuanced assessment of robustness. The authors validate their approach through experiments in adversarial learning and domain generalization scenarios, demonstrating that PA can effectively identify vulnerabilities in models that traditional metrics might overlook.

**Audience:**

Yes

**Broader Impact Concerns:**

I do not see any concerns on the ethical implications of this work.

**Claims And Evidence:**

Yes

**Requested Changes:**

- Please add more adversarial attack algorithms (e.g., AutoAttack) and perform the experiments on larger datasets like ImageNet.
- Please explain why the undefended model under Adversarial Ratio of 1.0 has such a high AFR.
- On CIFAR-10, any attack strength over $l_\infty$ of 8/255 may be too strong and makes the image unrecognizable to humans (but the AFR under 32 / 255 is still high, which looks weird to me).  Use ImageNet instead can solve this problem.
- Some notations are not clearly defined. For example, what is R(C, X)?

**Strengths And Weaknesses:**

Strengths
- The problem of assessing model robustness under covariate shift is important and interesting.
- The authors define the properties that a metric should have and identify the limitations of traditional metrics.
- The proposed PA metric is grounded in information theory and the Maximum Entropy Principle.

Weaknesses
- The experiments are inadequate. For instance, it only includes two attacks, PGD and FMN, and not the more advanced ones like AutoAttack. There is also only one small dataset of CIFAR-10 in this setting. Therefore, it is not convincing that the proposed metric can accurately reflect the robustness of models in real-world settings.
- Results in Figure 3 are strange. The undefended model under Adversarial Ratio of 1.0 should have very low AFR, but it is over 80%.
-  The proposed metric and the traditional metric AFR report similar trends and rankings, except that AFR tends to overestimate the robustness of the undefended model (which is questionable, as stated in my last point). This result makes the proposed metric less useful, because it is not very different from the existing ones.

---

> ### Author Response · Authors · 2025-08-19
> **Reply to review**
>
> We thank the reviewer for their feedback. In the following, we provide answers to their concerns and integrate these discussions into the document:
>
> **Use of AutoAttack:** Thank you for pointing this out. We replicated the adversarial scenario experiments with the AutoAttack library, testing the model with the APGD-CE/DLR/DLR$^\top$ and FAB attack on CIFAR-10 and Imagenet. The attacks are more effective, and AFR now ranges in 0-100%. As you pointed out, higher attack powers destroy the models' performance and are less indicative, so we also tested additional, lower attack powers. The general trends reported in our previous setting are confirmed: With $AFR_T$, the undefended model robustness is overestimated, and, additionally, the metric is less stable in its ranking. We have updated the experimental setting section in the main paper and included a new appendix, where we report all results.
>
> **$AFR_T$ vs PA:** Thanks for raising this point. Our paper questions the role of a performance metric ($AFR_T$) as a proxy for measuring robustness. We agree that in some cases $AFR_T$ displays similar behaviour to PA in the analysed scenarios. This is especially expected in settings with large shift magnitude. However, the metrics are substantially different. Below, we summarise in three points the reasons proposed in the paper.
>
> - *Controlled scenarios:* Fig 1 and Fig 10 describe a significant difference between PA and $AFR_T$ in controllable scenarios: With $AFR_T$, a constant classifier is less robust than a perfect one (fig. 1), and $AFR_T$ awards a model sensible to positive changes in the data as a more robust one (fig. 10). This is due to $AFR_T$ measuring performance and not robustness, different to PA, whose behavior is radically different in both examples.
>
> - *Adversarial Scenario:* Consider a binary classification scenario and assume a dataset ${X, Y}_i^N$. Let $Y’$, $Y’’$ be the model predictions respectively before and after an attack, with an adversarial ratio of $AR$, and let CR be the ratio of attacked observations that correctly changed their label. The post-attack $AFR_T$ can be then expressed as $AFR_T  = P(Y^{\prime\prime} = Y) = P(Y = Y^\prime) (1 - P(Y^{\prime\prime} \ne Y \mid Y^\prime = Y) + P(Y^{\prime\prime} = Y \mid Y^\prime \ne Y)) = P(Y = Y^\prime) + (1 - P(Y^\prime = Y))AR \cdot CR - P(Y^\prime = Y)AR \cdot CR = P(Y = Y^\prime) + AR \cdot CR (1 - 2 P(Y^\prime = Y))$. If a model has good classification performance, then $P(Y = Y^\prime) \rightarrow 1$, thus $AFR_T \approx 1 - AR \cdot CR$. If we consider a perfect attack with $CR = 1$, this entails that $AFR_T \approx 1 - AR$, therefore consistently over-scoring the robustness of a model. In the AFR_T plots, we can see that all models indeed score around the $1 - AR$ value. On the contrary, PA does not suffer from this phenomenon and detects an unrobust model immediately, even when the ratio of attacked observations is low. In addition to that, the analysis of the $\beta$ parameter provides further information on the models’ behaviour that cannot be made with $AFR_T$.
>
> - *Dom. Gen. Scenario:* In this case, the situation is even more heterogeneous. While $AFR_T$ continues to decrease monotonically with the severity of the shifts, PA highlights more fine-grained patterns. For example, in the DiagViB experiment, PA stabilizes once the model has exhausted the factors it actually represents (*e.g.*, hue and brightness), treating additional shifts as sampling noise. This behavior is meaningful: it indicates that the model is not adapting to new factors, even if accuracy remains superficially high. In contrast, $AFR_T$ conflates performance degradation with robustness, thereby masking these structural differences. This effect is visible in the comparison between ERM, IRM, and LISA, where $AFR_T$ suggests similar robustness, but PA reveals that IRM trades off invariance in low-shift regimes while LISA remains more stable across shifts. Thus, the two metrics tell complementary but non-equivalent stories, and only PA can expose these stability-related phenomena.

---

> > ### Author Response · Authors · 2025-08-19
> > **Reply to review (continue)**
> >
> > As a final word, we stress that we do not state to replace one metric with another, but we see PA as a more general robustness metric that should be preferred as the main metric in most cases. $AFR_T$ remains appropriate in settings where (i) sufficient labeled test samples are available, (ii) the full target domain is accessible, and (iii) the distribution shift is strong, since in this case AFR provides a simple and reliable estimate of test performance. However, if any of these assumptions fails, PA is better suited: (i) PA requires no supervised data, (ii) it provides robustness assessments that generalize to unseen domains (as shown in both adversarial and domain generalization experiments), and (iii) it is more discriminative under very mild shifts (e.g., in Appendix C, the Levenshtein IMDB experiment). This broader scope is the reason we emphasize PA as a more informative metric that should be complemented with the $AFR_T$ when applicable.
> >
> > **Notation unclear:** We have updated the notation to match the rest of the paper, without any loss of generality.

---

### Review · Reviewer_7tFM · 2025-06-28

**Summary Of Contributions:**

The paper proposes to use Posterior Agreement (PA) as a metric of robustness in scoring multi-class classifiers, applicable both for adversarial robustness and domain shift robustness, but also for model selection. The PA method was developed earlier (Buhmann 2010) but not used before in the context of robustness of multi-class classifiers. The empirical results demonstrate how the method adds more nuance in robustness analysis, compared to the commonly used Attack Failure Rate (AFR) metric.

**Audience:**

Yes

**Broader Impact Concerns:**

No concerns beyond the basic ethical implications of machine learning research.

**Claims And Evidence:**

Yes

**Requested Changes:**

Addressing all of the above weaknesses is critical for acceptance.

**Strengths And Weaknesses:**

Strengths:
* A well-justified novel use for the existing PA-measure.
* Detailed experiments and a strong analysis of results.
* Relevant theoretical results about the PA-measure.

Weaknesses:

There are some major weaknesses in explaining the content sufficiently clearly. The rest are relatively minor issues that can be fixed quite easily.

* Not clear whether PA is a metric in the sense of a metric space. I suggest to either demonstrate it by proving the triangle inequality or emphasize that the term metric is not used in this sense.

* The second property (shift-sensitive) on page 2 is quite vague. I understand that this may be partly intentional, but the vagueness should then be emphasized more clearly. What does it mean that generalization capabilities against covariate shift are the same or not for two models? And what does it mean that the metric is independent of the task performance? After reading the rest of the paper, I still feel that I don't have answers to these questions in the paper. It also seems to me that the paper does not explicitly claim that PA satisfies this Property 2. I suggest to change this property to be more precise, such that it would become possible to mathematically check whether a metric has this property.

* The description of PA in the introduction is not sufficiently understandable. Either make it better understandable or drop some of the text and stay at a more abstract level in the introduction. Currently, the sentence "The proposed PA metric provides a unique and unified framework for robustness assessment in the covariate shift setting, as it relies on a concept of robustness that does not stem from the nature of the data or the shift but instead from the consistency of the probabilistic response of the model" is not clear enough. I understand it after reading the whole paper, but it assumes too much from the reader at the early stage of the paper.

* It was not sufficiently clear from the text what the point of Fig. 1 is. I understood the overall sentiment, but not the full details.

* The paragraph starting with "Following epistemologically ..." at the bottom of page 3 is not clear enough, and due to this Fig. 2 remained unclear. Perhaps after reading Buhmann 2018 it would become clear, but the paper should be self-contained here. I understand what is meant by the datasets being 'with ideally identical signal, but different noise realizations' but the phrasing should be improved. However, the rest of the paragraph seems to assume that the reader should first read Buhmann 2018. In particular, it is not clear what the dataset X' and X'' are in Fig.2.

* I was expecting the Related Work section to include papers about PA also.

* More explanation should be added near Eq.(2). $F$ has been stated as fixed, and $c$ was defined through $f$ and $F$; hence, it was confusing to see $c$ as an argument of $R$ in Eq.(2), because it should be dependent on $F$. So in some sense, the reader should mentally drop $c=f\circ F$ before reading Eq.(2). I eventually understood what was meant, but it should be written more clearly.

* I suggest to explain the role of $\mu$ after (3e) in more detail. Currently, the meaning has to be put together by the reader from formulas and texts near Eq.(8) and Eq.(13a).

* The intuitive meaning of PA has not been sufficiently explained in the context when X' and X'' come from the same source. It has been briefly mentioned in the paragraph starting with "Following epistemologically" but this is too early for the reader to connect the dots. It is briefly stated also at the beginning of Section 3.2, but not in sufficient detail. I suggest to state the same explanation more clearly and in more detail somewhere in Section 3.2.

* The second paragraph in Section 4.2 starting with "In this scenario", is not clear enough. I didn't understand from this description how exactly the data were generated and what exactly $e_0^1,e_0^2,e_0,\dots,e_5$ meant. There are more details in the appendix, but the text in the main paper should still be clear, possibly referring to the appendix for some details, e.g. for the numbering of factors.

* The last paragraph of page 10 explains that for AR=0 the resulting beta is caused by particular properties of the Adam optimizer. Does it mean that using a different optimizer could give substantially different results? Perhaps this should be tested experimentally? If not testing other optimizers, then the paper should explicitly state the limitation that the method has been tested only on the Adam optimizer.

* The layout of Fig.7 could be improved by moving the third row of figures to the left, so that PA and AFR would align vertically. Maybe even better would be to have only two columns of figures, so that all different shift ratios for PA and AFR would align vertically, respectively.

* The choice of IRM and LISA was not justified. Why these and not some other methods?

* In my opinion, the claim that "PA is more suitable than AFR for evaluating robustness" on page 13 is overly strongly stated. As discussed later, AFR can reveal some details of shift 4 vs 5 which PA is not distinguishing and has a similar PA-value. I would say that PA and AFR provide different information and altogether both are valuable metrics of robustness.

* I didn't understand how exactly PA was used for model selection. What were X' and X'' in this case? Appendix E has more details, but the main paper should provide a sufficient explanation.

---

> ### Author Response · Authors · 2025-08-19
> **Reply to review**
>
> We thank the reviewer for their feedback. The questions raised helped us clarify several aspects of our work that were not properly described. In the following, we discuss them.
>
> **Is PA a metric:** Thank you for the question. We use the term metric in its generic sense, similarly to what is done with accuracy. Mathematically speaking, for a generic $\beta$, it is easy to prove that the empirical posterior agreement kernel $k$ (Eq. 14) is only a semi-metric since it does not satisfy the triangular inequality (in some sense, it measures a similarity). For example, let $X^\prime = (0.1, 0.9)$, $X^{\prime\prime} = (0.9, 0.1)$, $X^{\prime\prime\prime} = (0.5, 0.5)$, and $\beta = 1$, then
>
> $$
> k(X^\prime, X^{\prime\prime}) = \log(2 \cdot (0.1 \cdot 0.9 + 0.9 \cdot 0.1)) \approx -1.02
> $$
> $$
> k(X^{\prime\prime}, X^{\prime\prime\prime}) = \log(2 \cdot (0.9 \cdot 0.5 + 0.1 \cdot 0.5)) = 0
> $$
> $$
> k(X^\prime, X^{\prime\prime\prime}) = \log(2 \cdot (0.1 \cdot 0.5 + 0.9 \cdot 0.5)) = 0
> $$
>
> So $k(X^\prime, X^{\prime\prime})  + k(X^{\prime\prime}, X^{\prime\prime\prime}) < k(X^\prime, X^{\prime\prime\prime})$. Note that, even when considering $-k(X^\prime, X^{\prime\prime})$, the inequality does not, in general, hold.
>
> Similarly, we can prove that $PA(X^\prime, X^{\prime\prime})$ is a semi-metric but not a metric.
>
>
> **shift-sensitivity property is vague:** Yes, the absence of formalization is intentional, as a general description of concepts would require formalizing the concept of shift-power, which is hard or even impossible to do for more generic settings such as domain generalization. We have extended the discussion in the introduction to better highlight this problem. We have also provided a more detailed description of the two properties in the text, in terms of failure cases (i.e., what a robustness metric should not do).
>
> **Does PA comply with shift-sensitivity:** In short, an ideal robustness metric should respond only to covariate shift and not, for instance, to randomness induced by in-distribution sampling on an unperturbed dataset.  Designing such an ideal metric might be, however, impossible, as this would require insight into the data and noise generation process, which is often unknown. That said, PA better complies with shift-sensitivity than AFR. For example, a classifier that predicts the incorrect class but displays stable confidence in its predictions (i.e., the constant classifier, whose description has been updated) would score well with PA and very badly with AFR. We have updated the introduction to discuss this particular matter. Please, *cf.* also Rev iaqh, “$AFR_T$ vs PA”, for further technical details.
>
> **Description of PA is not understandable/does shift-sensitivity hold for PA/Paragraph “following epistemiologically…” not clear:** Thank you for pointing this out. We have taken the chance to rewrite that part in the introduction. We now introduce, in short, the original PA framework and its adaptation for robustness evaluation. We have, then, included a new “background” section in the methodology to discuss PA in more detail. Figure 2 has also been moved and introduced later.
>
> **Fig 1. not clear:** In the caption of the image, we have included further details about the three classifiers and the dataset used to perform the experiment.
>
> **Related work does not contain PA works:** We have updated the section, discussing several versions of the PA framework and applications to real-world problems.
>
> **Additional results obtained through Adam:** The information about the peakedness of the predictions is detectable because of the dampening effect of Adam in the optimization of $\beta$. In principle, this can be reached with other adaptive optimization algorithms such as AdaGrad, RMSProp, and Adam variants. We cannot guarantee that for non-adaptive optimization algorithms. We have further stressed in the text that the dampening is effective to obtain such an estimation.
>
> **Notation in Eq. 2 and what follows:** Thank you for pointing this out. Yes, in Eq. 2, $c$ is a simple index mapping, equivalent to the index set $\{c(x_1), \dots, c(x_N)\}$, for $x \in D_X$. This is different from the more general $c = f \circ F$ that works on $\mathcal{X}$. Our focus here is to devise a surrogate posterior over $\Theta$, and both definitions induce a partition on $\Theta \ni \theta$. This represents an interesting point for future study. We have updated the footnote with more explanation.
>
>
> **Alignment in Fig. 7:**  We have updated the figure according to the suggestion.

---

> > ### Author Response · Authors · 2025-08-19
> > **Reply to review (continue)**
> >
> > **Description of data generation for DG:** We have clarified the description in Section 4.2. In particular, we now explicitly define what we mean by *factors* (texture, hue, lightness, position, scale) and by *domains* (specific configurations of these factors). We also explain the notation: $e_0^1$ and $e_0^2$ are the two training domains with disjoint factor values, validation domains replicate the same configurations with disjoint MNIST samples, and the six test domains $\{e_0, e_1, \dots, e_5\}$ are constructed by progressively modifying 0 to 5 factors relative to $e_0^1$. This step-by-step clarification makes the data generation process self-contained in the main text, while the appendix continues to provide full details of the factor numbering.
> >
> > **Method choice for DG experiment:** We aimed to showcase the breadth of applicability of PA by including two archetypal approaches to domain generalization. IRM enforces invariance in the latent representation space, while LISA instead acts directly in the data space through selective augmentation. By contrasting these two pathways, we emphasize that PA can capture robustness properties across fundamentally different methodological families, beyond simple ERM baselines. This rationale has been added to the main text.
> >
> > **Strength of claim “PA more suitable than AFR”:** Thank you for pointing this out. We have softened the claim in Section 4.2. Our intention is not to dismiss AFR, but to clarify that PA and AFR provide complementary perspectives on robustness. While AFR highlights performance variations across shifts, PA captures changes in predictive distributions that AFR may overlook (*e.g.*, stabilization after hue and brightness shifts). *cf.* also Rev iaqh “$AFR_T$ vs PA”. We now frame the two metrics as jointly informative in the context of domain generalization.
> >
> >
> > **Model selection experiment (clarity of $X^\prime$ and $X^{\prime\prime}$):** We have expanded Section 4.3 to clarify how PA is applied for model selection. In particular, we now explicitly explain that training uses datasets $X^\prime$ and $X^{\prime\prime}$, which differ only in the hue factor, and that these datasets are resampled for each shortcut-opportunity configuration. Validation is carried out on matched splits that follow the same hue-based variation. Epoch-wise model selection is then performed by maximizing either validation accuracy or validation PA, and the resulting generalization performance under cumulative shifts is compared. This explanation has been added before referring the reader to Appendix E for the full setup details.

---

### Review · Reviewer_c1ic · 2025-07-29

**Summary Of Contributions:**

The paper introduces a robustness evaluation metric named Posterior Agreement (PA), specifically designed to assess the robustness of machine learning models under covariate shifts. The PA measures the consistency in model predictions across perturbed and original datasets. The authors demonstrate theoretically and empirically that PA satisfies essential properties for robustness evaluation, including shift sensitivity and non-increasing.

**Audience:**

Yes

**Broader Impact Concerns:**

Not apply

**Claims And Evidence:**

Yes

**Requested Changes:**

1. More experiments on ImageNet, which is closer to a real-world scenario, would make the paper stronger.
2. Maybe I missed it, but could the author discuss more about the parameter $\beta$? specifically from the computational cost point of view?

**Strengths And Weaknesses:**

Strengths

1. The metric seems novel; PA provides a theoretically sound robustness evaluation.
2. The paper is well written.
3. The paper did the experiments with PA across adversarial and domain generalization contexts, demonstrating its power.

Weakness

1. The experiment is only conducted with CIFAR-10, which is less practical.
2. It seems like the $\beta$ is an very important parameter which impacts the results, and it is computational expensive to search for the critical parameter, which makes the PA less practical

---

> ### Author Response · Authors · 2025-08-19
> **Reply to review**
>
> We thank the reviewer for their comments. We have addressed them and we discuss them in the following.
>
> **Experiments with ImageNet:** We have updated the experiments, including ImageNet. The paper now contains additional discussion in the experimental settings section and an additional appendix. The results are similar to those attested in CIFAR-10, as expected by the behavior of the metric. Please, refer also to Rev iaqh par. “use of autoattack”.
>
> **$\beta$ computational time:** The optimization is conducted with a standard PyTorch routine (*cf.* also the codebase). The optimization is performed with Adam (albeit other optimizers could be used) over a single scalar only, where the dataset is constituted by the logits of the model (which can also be stored, in case of recomputation of the metric). Therefore, due to the low complexity (single minimum) and dimensionality (single scalar), the optimization problem can be very efficiently solved, in particular for both CIFAR-10 and ImageNet, it runs in the order of tens of minutes with a single GPU acceleration. In any case, we believe that the computational time does not constitute a severe problem, since we conduct evaluation experiments (except for model selection), which can be considered offline studies, and do not have, therefore, stringent time requirements. We have included a consideration in the main paper.

---

### Decision · Action_Editor_Dqmi · 2025-09-16

**Recommendation:** Accept as is

**Audience:**

Yes

**Audience Explanation:**

The paper addresses an important topic of measuring robustness in machine learning, and many researchers might be interested in this topic.

**Claims And Evidence:**

Yes

**Claims Explanation:**

The paper proposes a new metric focused on robustness, rather than prediction accuracy. The metric is based on Posterior Agreement which is a framework previously proposed for other purposes. Here, the metric is applied to measure robustness of machine learning models. The approach is simple, even though it takes sometime to understand how it works, and one reviewer explicitly mentioned that the readability can be improved. I agree to some extent: it took me some time to figure out how X and X' were supposed to be designed why measuring the overlap with the temperature parameter could be insightful. Even after this I am not entirely sure how I would use the method. I encourage authors to improve this aspect to increase the impact of their work.

The reviewers were mostly satisfied with the rebuttal, with some concerns about additional experiments that remained. I believe the authors have tried enough to do their best and I think it is okay to let the paper be published in its current form.